# CONTEXTOR: Contextualized High-order Contrastive Learning

Ze Cai [1 2 *]   Hanzhe Liang [3 2 *]   Sihang Zeng [4]   Binbin Zhou [1 †]   Jun Wen [2 †]

## Abstract

High-order relations involving multiple interacting entities are commonly encountered, particularly in biomedical domains. Existing relational learning methods typically learn static entity representations and assume symmetric relation inference, which can be inadequate for capturing context-dependent entity functions and the inherent asymmetry of high-order relations. In this paper, we propose Contextualized High-order Contrastive Learning (CONTEXTOR), a general-purpose, plug-and-play framework that formulates high-order relation inference as a dynamic query–response process. Specifically, CONTEXTOR decomposes each high-order relation into multiple incomplete query tuples and their corresponding response entities. Given a query tuple, we contextualize candidate response entity representations via an asymmetric conditional modulation, and align queries with their corresponding contextualized responses through multi-fold contrastive learning. Extensive experiments on benchmark datasets spanning multiple biomedical tasks demonstrate that CONTEXTOR consistently outperforms existing methods across diverse evaluation settings. Code is available at https://github.com/ZJUDataIntelligence/CONTEXTOR.

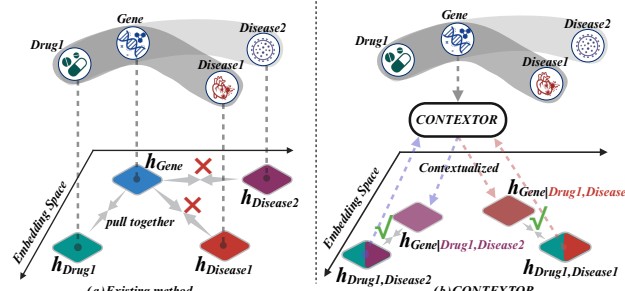

*Figure 1.* Comparison between (a) existing methods and (b) CONTEXTOR. Existing methods learn a single static embedding for each entity, which can conflate distinct semantic roles when the same entity (e.g., the *gene* in the figure) appears in different relational contexts (e.g., *disease1* and *disease2*). In contrast, CONTEXTOR produces context-aware entity representations, enabling flexible and asymmetric high-order relation inference.

## 1. Introduction

High-order relation inference is increasingly encountered in real-world applications, particularly in computational biology and translational medicine (Peng et al., 2017; Chiang et al., 2018; Zeng et al., 2020), such as drug–gene–disease associations in drug development and drug–drug–cell line interactions underlying synergistic cancer therapies. However, these relations inherently involve multiple interacting entities, giving rise to a combinatorial explosion in the space of possible relational tuples (Liu et al., 2022; Wu et al., 2025a). Moreover, biomedical entities often exhibit strong context-dependent functional roles, causing severe semantic ambiguity when the same entity participates in different relational compositions. This further renders comprehensive annotation through wet-lab experiments or expert curation prohibitively expensive. As a result, existing high-order relation datasets are typically sparse, incomplete, and biased toward well-studied entity combinations. Consequently, there has been growing interest in leveraging computational approaches, particularly machine learning–based methods, to automate the discovery and inference of high-order relations.

In contrast to traditional relation extraction methods (Öztürk et al., 2018; Preuer et al., 2018), contrastive learning (CL) has emerged as a powerful paradigm for both representation learning (Chen et al., 2020) and relational inference (Jia et al., 2026). By explicitly contrasting positive and negative samples, CL-based approaches have demonstrated improved label efficiency across a range of biomedical applications (Chen et al., 2025; Qin et al., 2024; Jiang et al., 2025;

---

[*]Equal contribution [1]School of Computer and Computing Science, Hangzhou City University, China [2]Biological and Life Sciences Division, Mohamed bin Zayed University of Artificial Intelligence, UAE [3]Shenzhen Audencia Financial Technology Institute, Shenzhen University, China [4]Department of Biomedical Informatics and Medical Education, University of Washington, US. Correspondence to: Jun Wen <jun.wen@mbzuai.ac.ae>, Binbin Zhou <bbzhou@hzcu.edu.cn>.

*Proceedings of the 43rd International Conference on Machine Learning*, Seoul, South Korea. PMLR 306, 2026. Copyright 2026 by the author(s).

Zeng et al., 2025), including drug–target binding (Jia et al., 2026) and drug synergy prediction (Luo et al., 2025). Beyond enhanced label efficiency, contrastive representations have also been shown to generalize effectively to previously unseen entities, facilitating robust relation inference under data-sparse settings (Jia et al., 2026).

Despite these advances, existing methods still face limitations in modeling high-order biomedical interactions. The key challenges are summarized as follows. **1) Contextualized entity functions:** Most existing approaches rely on static entity representations, implicitly assuming fixed functions or semantics, as illustrated in Figure 1. This assumption is incompatible with biomedical systems, where an entity's functional role is highly conditional on its interaction context. For example, the antiplatelet drug clopidogrel induces adverse cardiovascular outcomes primarily in patients carrying loss-of-function variants in gene *CYP2C19*, while exhibiting normal efficacy in other genetic backgrounds; similarly, drug synergy often emerges only under specific cellular states. Such context-dependent behaviors force a single static embedding to conflate multiple functional roles, leading to semantic entanglement across different relational compositions. **2) Asymmetric high-order relations:** Existing contrastive learning methods predominantly focus on pairwise relations (e.g., text–image or drug–protein), which are largely symmetric by design. In contrast, high-order biomedical relations are inherently asymmetric. For instance, predicting an adverse drug reaction given a drug–gene context constitutes a fundamentally different inference task from identifying an associated gene given the observed drug-induced adverse event. As a result, existing methods fail to flexibly capture the heterogeneous conditional dependencies.

In this paper, we propose **Contextualized High-order Contrastive Learning (CONTEXTOR)**, a plug-and-play contrastive learning framework for high-order relation inference. CONTEXTOR formulates high-order relation inference as a dynamic query–response process, decomposing each observed high-order relation into multiple incomplete query tuples and corresponding response entities. Given a query tuple, CONTEXTOR contextualizes candidate responses via an Asymmetric Conditional Modulation (ACM) module, which generates query-dependent modulation parameters to apply directional and context-aware transformations to candidate embeddings. CONTEXTOR then aligns queries with their corresponding contextualized responses through contrastive learning, enabling asymmetric, context-sensitive inference under sparse and combinatorial relational settings. Our key contributions are summarized as follows:

- We propose CONTEXTOR, a novel contrastive learning framework explicitly designed for high-order relation inference, which is modular and adaptable to diverse backbone architectures, including MLPs and graph neural networks.

- We introduce an *asymmetric conditional modulation* module together with a query-response contrastive learning strategy to enable directional and context-sensitive inference, addressing the asymmetry and semantic polysemy inherent in high-order biomedical relations.

- We introduce a new benchmark dataset, which comprises Drug–Gene–Adverse Drug Reaction (ADR) relations, to facilitate systematic evaluation of high-order relational modeling.

- Extensive experiments on three benchmark datasets, including Drug–Microbe–Disease, Drug Synergy, and Drug–Gene–ADR, demonstrate that CONTEXTOR consistently outperforms state-of-the-art baselines across diverse evaluation settings.

## 2. Related Work

### 2.1. Contrastive Representation Learning

Self-supervised contrastive learning (CL) has emerged as a dominant paradigm for representation learning, with seminal frameworks such as SimCLR (Chen et al., 2020) and MoCo (He et al., 2020) emphasizing augmentation-invariant and globally aligned embeddings. This paradigm has been widely adopted in biomedical applications. For example, MolCLR (Wang et al., 2022) learns invariant molecular representations via graph augmentations, while CCL-ASPS (Tian et al., 2024), CSCL-DTI (Lin et al., 2024), and MCL-DDI (Li et al., 2025) employ contrastive objectives to align biological entities across multiple views or modalities.

Despite their success, these methods fundamentally learn static and context-independent representations, implicitly assuming that a biological entity maintains a fixed semantic role, as adopted by methods such as HNCL-DTI (Hu et al., 2024), KnowMDD (Lin et al., 2025), and MCHNN (Liu et al., 2023). While effective for tasks requiring a single robust embedding per entity, such invariant representations are less suitable for high-order, context-dependent relations, where an entity's meaning is jointly determined by co-occurring entities. Moreover, standard CL frameworks rely on symmetric similarity between fixed embeddings, limiting their ability to model the asymmetric and conditional nature of query–response inference.

### 2.2. Contextual Representation Learning

Standard representation learning often relies on static embeddings, which fail to capture the conditional and role-dependent nature of entity interactions. Recent work addresses this limitation through context-aware representa-

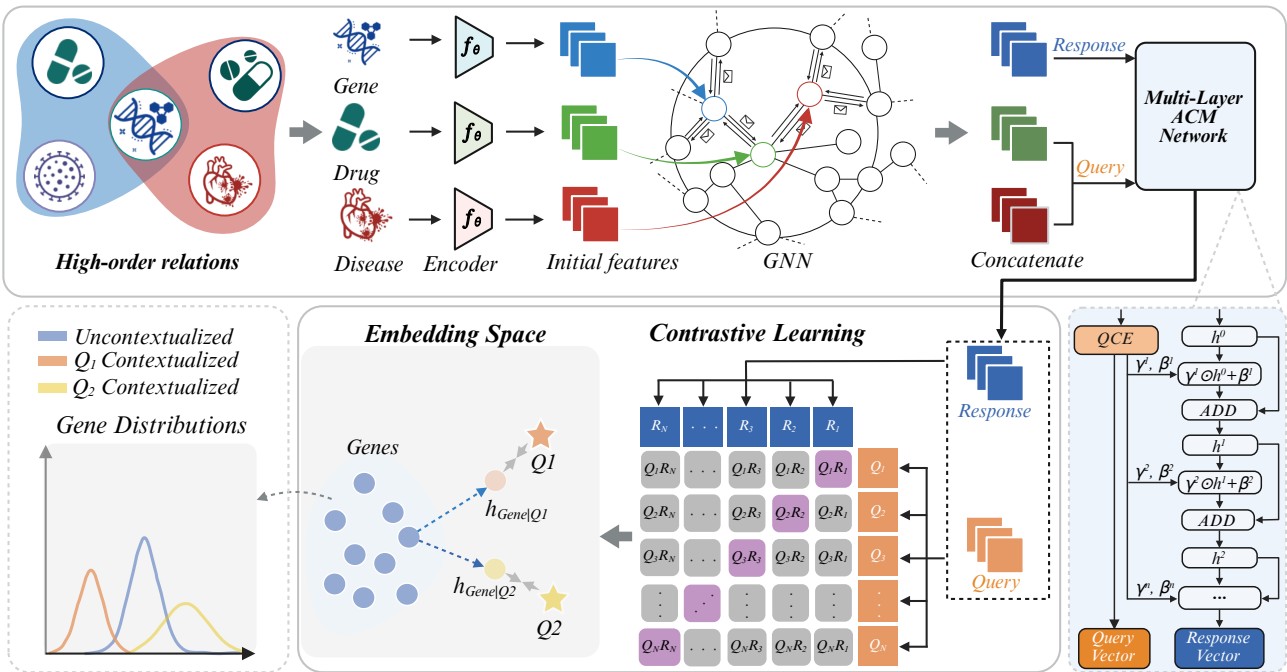

*Figure 2.* Architecture of the proposed **CONTEXTOR framework**, which is designed to be plug-and-play and consists of two main components: (1) an **Asymmetric Conditional Modulation (ACM)** module that dynamically conditions candidate entity representations on query-specific context; and (2) a **Contrastive Query–response Learning** objective that aligns queries with their corresponding conditional responses under asymmetric inference settings.

tions that adapt to queries or environments at inference time. In the broader machine learning literature, such approaches generally model context through input structuring or inference-time adaptation, as demonstrated in studies on latent structure tracing (Park et al., 2024), structured prompting (Huang et al., 2023), prompt selection and temporal causality (Wu et al., 2025b; Cheng et al., 2025), and context-driven evaluation protocols (Rauba et al., 2024).

Despite these efforts, existing context-aware and contrastive learning methods remain insufficient for high-order relational reasoning. Most existing methods focus on single entities or pairs, treating context implicitly rather than explicitly conditioning on high-order relational queries or inference direction. In high-order relations, an entity's semantic role is jointly determined by multiple interacting entities, making static or globally shared embeddings prone to semantic entanglement and poor generalization in combinatorial, sparsely observed regimes. This limitation is especially pronounced in biomedical applications. While prior work has incorporated contextual information in drug–drug interaction prediction (Lu et al., 2024), modeled high-order dependencies or dynamic protein representations (Lu et al., 2025; Sicherman & Radinsky, 2025), and leveraged global context in spatial transcriptomics (Oh et al., 2025), these methods remain largely task-specific or grounded in pairwise objectives, and do not provide a general mechanism for conditioning entity representations on high-order relations.

## 3. Preliminaries

In this section, we first outline the core concepts and problem formulation of CONTEXTOR. All symbols and their definitions are provided in the *Supplementary Material A*.

### 3.1. Definitions

**Definition 3.1.1** (Query-Conditioned Relational Tuple). Let $\mathcal{E}_1, \ldots, \mathcal{E}_m$ denote the sets of entities corresponding to each role in the relation. A query-conditioned relational tuple is defined as an ordered tuple

$$\mathbf{e} = (e_1, \ldots, e_m) \in \mathcal{E}_1 \times \cdots \times \mathcal{E}_m. \quad (1)$$

Given such a tuple, one entity $e_k$ with $k \in \{1, \ldots, m\}$ is treated as unobserved, while the remaining entities are observed and used to construct the query.

**Definition 3.1.2** (Query Function). Given a query-conditioned relational tuple $\mathbf{e} = (e_1, \ldots, e_m)$ with $e_k$ unobserved for some $k \in \{1, \ldots, m\}$, let $\mathbf{h}_{i_1}, \ldots, \mathbf{h}_{i_{m-1}} \in \mathbb{R}^d$ denote the embeddings of the observed entities. The query function $\mathcal{Q} : (\mathbb{R}^d)^{m-1} \to \mathbb{R}^c$ is defined as

$$\mathbf{z}_q = \mathcal{Q}(\mathbf{h}_{i_1}, \ldots, \mathbf{h}_{i_{m-1}}). \quad (2)$$

**Definition 3.1.3** (Asymmetric Conditional Modulation Operator). Given a query vector $\mathbf{z}_q \in \mathbb{R}^c$, the asymmetric conditional modulation operator is defined as a query-

conditioned mapping

$$\mathcal{M}(\cdot; \mathbf{z}_q) : \mathbb{R}^d \to \mathbb{R}^d, \tag{3}$$

which maps a base entity embedding $\mathbf{h}_j$ to a modulated embedding $\widetilde{\mathbf{h}}_j = \mathcal{M}(\mathbf{h}_j; \mathbf{z}_q)$.

**Definition 3.1.4** (Conditional Entity Embedding). Given a query vector $\mathbf{z}_q \in \mathbb{R}^c$ and a base entity embedding $\mathbf{h}_j \in \mathbb{R}^d$, the conditional entity embedding is defined as

$$\mathbf{h}'_j = \mathcal{P}(\mathcal{M}(\mathbf{h}_j; \mathbf{z}_q)) \in \mathbb{R}^c, \tag{4}$$

where $\mathcal{P} : \mathbb{R}^d \to \mathbb{R}^c$.

**Definition 3.1.5** (Query–Response Scoring Function). The query–response scoring function is defined as

$$s : \mathbb{R}^c \times \mathbb{R}^c \to \mathbb{R}. \tag{5}$$

## 3.2. Problem Statement

Given a query-conditioned relational tuple $\mathbf{e} = (e_1, \dots, e_m)$ with $e_k$ unobserved for some $k \in \{1, \dots, m\}$, and base embeddings of the observed entities, a query representation $\mathbf{z}_q$ is constructed via the query function $\mathcal{Q}$. The learning objective is to infer the unobserved entity by ranking candidate entities based on their query-conditioned compatibility, where each candidate is represented by a conditional entity embedding $\mathbf{h}' = \mathcal{P}(\mathcal{M}(\mathbf{h}; \mathbf{z}_q))$ and scored by the query-response function $s(\mathbf{z}_q, \mathbf{h}')$.

# 4. Methodology

CONTEXTOR comprises a Query-Conditioned Encoder (QCE), which integrates an Asymmetric Conditional Modulation module with a Contrastive Query–response Learning strategy. An overview is illustrated in Figure 2.

## 4.1. Query-Conditioned Encoders

For a query-conditioned relational tuple $\mathbf{e} = (e_1, \dots, e_m)$ with $e_k$ unobserved, the query is constructed from the remaining $m - 1$ observed entities to encode the relational context. In this work, we instantiate this formulation with triplets ($m = 3$), where the query representation is obtained by fusing the embeddings of the two observed entities. Let $\mathbf{h}_u, \mathbf{h}_v \in \mathbb{R}^d$ denote their base embeddings. The query vector is computed as

$$\mathbf{z}_q = \mathrm{MLP}_{\mathrm{qry}}([\mathbf{h}_u; \mathbf{h}_v]). \tag{6}$$

where $[\cdot; \cdot]$ denotes concatenation and $\mathbf{z}_q \in \mathbb{R}^c$ encodes the query-specific relational context.

Under this relational context, an entity's representation should vary accordingly rather than remain static. We therefore condition candidate entity embeddings on the query

vector via an **Asymmetric Conditional Modulation** (ACM) mechanism. For a candidate entity $e_j$ with base embedding $\mathbf{h}_j \in \mathbb{R}^d$, ACM applies a query-dependent feature-wise affine transformation,

$$\mathcal{M}(\mathbf{h}_j; \mathbf{z}_q) = \gamma_q \odot \mathbf{h}_j + \beta_q, \tag{7}$$

where the modulation parameters $\gamma_q, \beta_q \in \mathbb{R}^d$ are generated from the query representation,

$$\gamma_q = W_\gamma \mathbf{z}_q + \mathbf{b}_\gamma, \quad \beta_q = W_\beta \mathbf{z}_q + \mathbf{b}_\beta. \tag{8}$$

The conditional transformation is further refined by stacking multiple ACM layers with residual connections. The $\ell$-th layer updates the candidate representation as

$$\mathbf{h}_j^{(\ell)} = \mathbf{h}_j^{(\ell-1)} + \gamma_q^{(\ell)} \odot \mathbf{h}_j^{(\ell-1)} + \beta_q^{(\ell)}, \tag{9}$$

where $\mathbf{h}_j^{(0)} = \mathbf{h}_j$ and $\gamma_q^{(\ell)}, \beta_q^{(\ell)}$ are computed from $\mathbf{z}_q$ using layer-specific parameters. The output of the final layer is projected into the query space to obtain the conditional entity embedding,

$$\mathbf{h}'_j = \mathrm{MLP}_{\mathrm{proj}}(\mathbf{h}_j^{(L)}) \in \mathbb{R}^c. \tag{10}$$

## 4.2. Contrastive Query-response Learning

To effectively model the conditional dependencies among entities in high-order biomedical relationships, our objective is to organize the latent space such that the query embedding $\mathbf{z}_q$ is tightly aligned with the conditional representation of the correct response $\mathbf{h}'_j$, while ensuring clear separation from irrelevant (negative) candidates. This alignment promotes the learning of more discriminative and context-aware embeddings.

Given a minibatch of $B$ positive query–response pairs, we construct negative samples for each query by considering the responses from the other query–response pairs in the batch as contrasting candidates. For the $i$-th query embedding $\mathbf{z}_{q_i}$ and its corresponding positive response $\mathbf{h}'_{j_i}$, we employ the InfoNCE loss (Oord et al., 2018) to optimize the alignment of positive samples while pushing negative samples apart in the embedding space:

$$\mathcal{L}_i = -\log \frac{\exp\left(\mathrm{sim}(\mathbf{z}_{q_i}, \mathbf{h}'_{j_i})/\tau\right)}{\sum_{l=1}^{B} \exp\left(\mathrm{sim}(\mathbf{z}_{q_i}, \mathbf{h}'_{j_l})/\tau\right)}, \tag{11}$$

where $\tau$ is a temperature hyperparameter and $\mathrm{sim}(\cdot, \cdot)$ denote cosine similarity.

## 4.3. Overall Training Objective

To enable inference over any unobserved entity in a relational tuple $\mathbf{e} = (e_1, \dots, e_m)$, we consider all query configurations corresponding to different choices of the unobserved entity. Formally, for $k \in \{1, \dots, m\}$, the $k$-th query

| Methods | Hits@K (%) | | | NDCG@K (%) | | |
|---|---|---|---|---|---|---|
| | @1 | @3 | @5 | @1 | @3 | @5 |
| RF | 34.06 | 57.28 | 68.31 | 34.06 | 47.51 | 52.08 |
| MLP | 42.72 | 65.40 | 73.15 | 42.72 | 55.88 | 59.99 |
| CP | 44.73 | 66.74 | 76.24 | 44.73 | 57.50 | 61.51 |
| Tucker | 45.27 | 67.54 | 76.98 | 45.27 | 61.62 | 63.90 |
| CoSTCo | 38.69 | 60.38 | 71.77 | 38.69 | 51.21 | 55.93 |
| GCN | 62.66 | 76.57 | 79.74 | 62.66 | 71.00 | 72.31 |
| GraphSAGE | 56.98 | 73.52 | 77.42 | 56.98 | 66.84 | 68.45 |
| GAT | 47.13 | 69.60 | 73.96 | 47.13 | 59.35 | 62.38 |
| GIN | 40.08 | 60.69 | 69.29 | 40.08 | 52.27 | 57.15 |
| DHNE | 81.86 | 93.66 | 96.02 | 81.86 | 88.88 | 89.86 |
| HyperSAGNN | 87.04 | 94.51 | 96.13 | 87.04 | 91.18 | 92.05 |
| HGSynergy | 88.68 | 90.99 | 94.51 | 88.68 | 90.82 | 91.92 |
| MCHNN | 90.04 | 93.92 | 95.41 | 90.04 | 92.38 | 92.91 |
| **CONTEXTOR** | **93.30**$_{\pm 2.09}$ | **95.77**$_{\pm 1.61}$ | **96.51**$_{\pm 1.79}$ | **93.30**$_{\pm 2.09}$ | **94.74**$_{\pm 1.50}$ | **95.05**$_{\pm 1.73}$ |

*Table 1.* Performance comparison on the DMD dataset. The results are averaged from the four scenarios. The best result for each metric is highlighted in **bold**, and the second-best is underlined.

configuration is defined as

$$q^{(k)} : \mathbf{e} = (e_1, \ldots, e_m), \quad e_k \text{ unobserved.} \quad (12)$$

For each query configuration $q^{(k)}$, we follow a unified pipeline: (i) construct a query vector $\mathbf{z}_q$ from the observed entities, (ii) modulate candidate embeddings to obtain conditional representations $\mathbf{h}'$, and (iii) compute a contrastive InfoNCE loss (Eq. (11)). Let $\mathcal{L}^{(k)}$ denote the contrastive loss associated with configuration $q^{(k)}$. The overall training objective is defined by averaging the losses across all configurations,

$$\mathcal{L}_{\text{overall}} = \frac{1}{m} \sum_{k=1}^{m} \frac{1}{B} \sum_{i=1}^{B} \left[ -\frac{\text{sim}(\mathbf{z}_{q_i}^{(k)}, \mathbf{h}_{j_i}^{\prime(k)})}{\tau} \right. $$
$$\left. + \log \left( \sum_{l=1}^{B} \exp \left( \frac{\text{sim}(\mathbf{z}_{q_i}^{(k)}, \mathbf{h}_{j_l}^{\prime(k)})}{\tau} \right) \right) \right]. \quad (13)$$

This multi-query approach enables the model to reason about any missing entity in a triple, making it more flexible and suitable for a wide range of downstream tasks, such as knowledge completion, drug discovery, and hypothesis generation. By integrating the loss across all query configurations, we ensure that the model learns consistent, context-aware entity representations.

We show that CONTEXTOR alleviates the geometric bottleneck of static embeddings under semantic polysemy and enables context-aware optimization through ACM. We further interpret our contrastive objective as maximizing a variational lower bound on the mutual information between queries and query-conditioned representations. Formal derivations are provided in the *Supplementary Material B*.

## 5. Experiments

### 5.1. Implementation

#### 5.1.1. DATASETS

We conduct experiments on three biologically meaningful entity association datasets: DMD (Drug–Microbe–Disease), Drug Synergy, and DGA (Drug–Gene–ADR). (1) **DMD dataset.** It contains 2,763 drug–microbe–disease triplets collected from MDAD (Sun et al., 2018) and HMDAD (Ma et al., 2017), involving 270 drugs, 58 microbes, and 167 diseases. Drug features are derived from SMILES (Pub-Chem (Kim et al., 2019)), while microbe and disease features are based on NCBI Taxonomy and MeSH. (2) **Drug Synergy dataset.** This dataset is collected from NCI-ALMANAC (Holbeck et al., 2017) and O'Neil (O'Neil et al., 2016) and consists of drug–drug–cell-line triplets with synergy scores. The final datasets include 74,139 triplets (87 drugs, 55 cell lines) and 18,950 triplets (38 drugs, 39 cell lines), respectively. (3) **DGA dataset.** It is constructed by integrating PharmGKB (Gong et al., 2021) and DrugBank (Knox et al., 2024) and comprises 1,270 drug–gene–ADR triplets covering 217 drugs, 384 genes, and 180 ADRs. ADRs are standardized using UMLS CUIs (Bo-denreider, 2004). Detailed information about the datasets is presented in the *Supplementary Materials C.1*.

#### 5.1.2. BASELINES

Following prior work, we evaluate our method on the DMD, Drug Synergy, and DGA datasets against representative baselines. (1) **DMD dataset.** Following MCHNN (Liu et al., 2023), we compare with non-graph-based methods (RF, MLP, CP, Tucker, CoSTCo), graph neural networks (GCN (Kipf, 2016), GraphSAGE, GAT, GIN), and hypergraph-based models (DHNE (Tu et al., 2018), Hy-perSAGNN (Zhang et al., 2019), HGSynergy (Liu et al.,

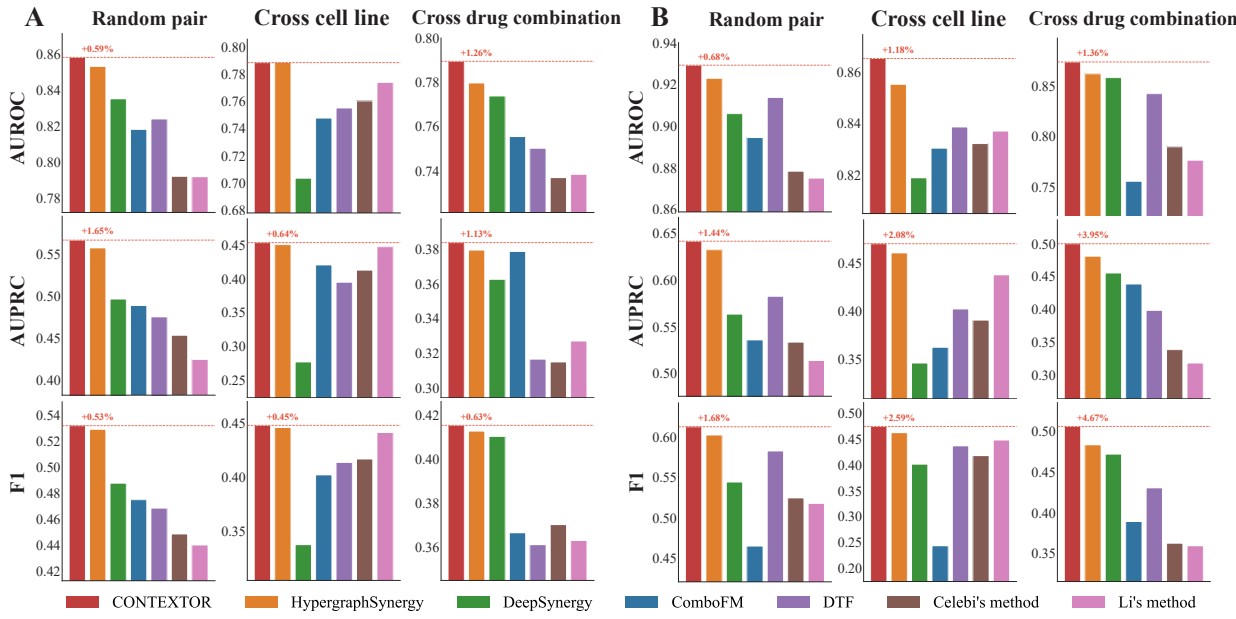

*Figure 3.* Performance of drug synergy prediction on the **(A) ALMANAC dataset** and **(B) O'Neil dataset** with three different evaluation settings. The detailed numerical results are provided in the *Supplementary Materials C.4*.

2022), MCHNN (Liu et al., 2023)). (2) **Drug Synergy dataset.** Following HGSynergy (Liu et al., 2022), we compare CONTEXTOR with DeepSynergy (Preuer et al., 2018), DTF (Sun et al., 2020), ComboFM (Julkunen et al., 2020), Celebi's method (Celebi et al., 2019), Li's method (Li et al., 2018), and HGSynergy (Liu et al., 2022). (3) **DGA dataset.** We evaluate CONTEXTOR under different baseline models, including MLP, GCN, and HGNN (Feng et al., 2019). Baseline selection and implementation details are provided in the *Supplementary Material C.2*.

### 5.1.3. EVALUATION METRICS

We adopt dataset-specific evaluation metrics following prior work. **(1) DMD dataset.** We report Hits@1/3/5 and normalized discounted cumulative gain at 1/3/5 (NDCG@1/3/5) following MCHNN. **(2) Drug Synergy dataset.** We report Area Under the Receiver Operator Curve (AUROC), Area Under the Precision–Recall Curve (AUPRC), and F1-score, consistent with previous synergy prediction studies. **(3) DGA dataset.** We evaluate performance using AUROC, AUPRC, and mean reciprocal ranking (MRR). The rationale for metric selection and detailed computation procedures are provided in the *Supplementary Material C.3*.

### 5.1.4. EXPERIMENTAL DETAILS

For the DMD and Drug Synergy datasets, we strictly follow the experimental protocols of MCHNN (Liu et al., 2023) and HGSynergy (Liu et al., 2022), respectively. To ensure a fair comparison, all model architectures, data splits, hyperparameter settings, and evaluation metrics are kept identical

to the original works, except that the original loss functions are replaced with our proposed CONTEXTOR objective. Specifically, for the DMD dataset, we substitute the DGI-based contrastive loss used in MCHNN with CONTEXTOR, while preserving the hierarchical message-passing mechanism and training strategy. For the Drug Synergy dataset, the reconstruction loss adopted in HGSynergy is replaced by CONTEXTOR, with all other components and optimization settings unchanged.

The DGA dataset comprises ternary drug–gene–ADR associations, which naturally support a range of biomedical investigation scenarios. Specifically, we consider three representative tasks: (a) **Drug discovery**, where drug candidates are screened given a gene and associated ADRs; (b) **Gene identification**, where causal genes are inferred for a given drug with observed ADRs; and (c) **ADR prediction**, where potential adverse drug reactions are predicted given a drug and its related gene effects. We train each architecture in both a vanilla setting using binary cross-entropy (BCE) loss and a CONTEXTOR-augmented setting using a combined BCE and contrastive loss. Detailed training and implementation configurations can be found in the *Supplementary Materials C.2.4*.

## 5.2. Main Results

### 5.2.1. PERFORMANCE ON DMD DATASET

Table 1 presents the performance comparison on the DMD dataset. It is evident that the model equipped with our proposed contrastive learning module consistently outper-

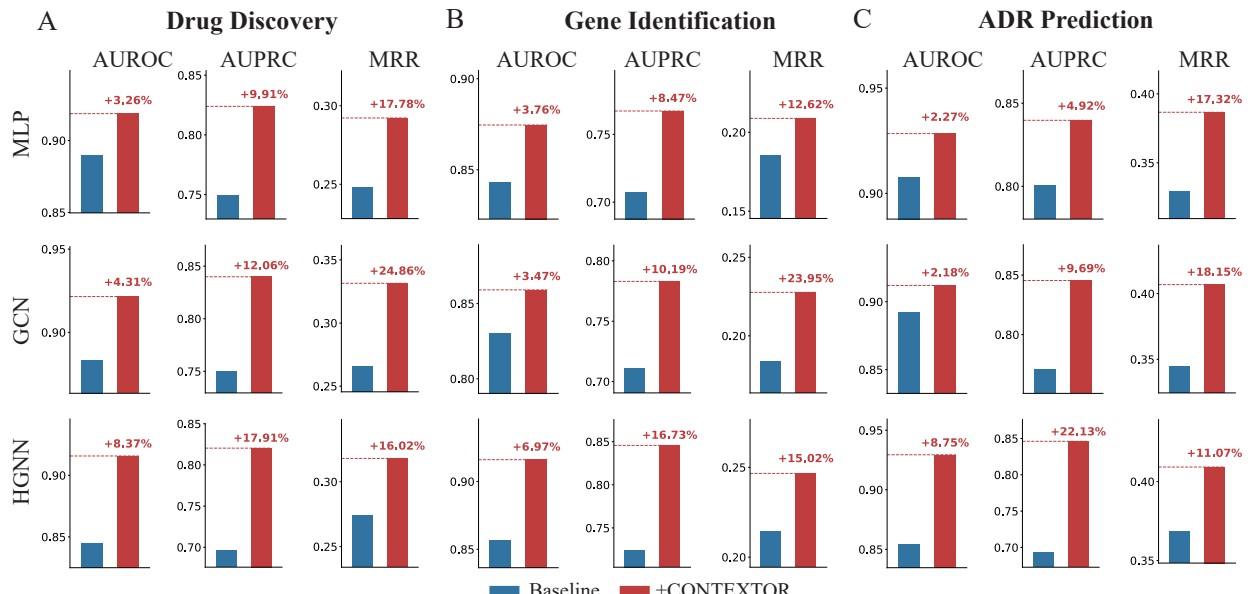

*Figure 4.* Model performance on the DGA dataset with three different evaluation settings: **(A) Drug Discovery**, **(B) Gene Identification**, and **(C) ADR Prediction**.

forms all baselines across all metrics. Specifically, it surpasses the strongest baseline by +3.26%, +1.26%, and +0.38% on Hits@1, Hits@3, and Hits@5, respectively. Similar improvements are also observed in NDCG metrics, demonstrating the robustness of the proposed module under top-k retrieval evaluation. We also observe that hypergraph-based models significantly outperform graph-and non-graph-based methods, underscoring the advantage of capturing high-order relations among drugs, microbes, and diseases. These results demonstrate the effectiveness of our contrastive learning module in injecting context into static entity embeddings.

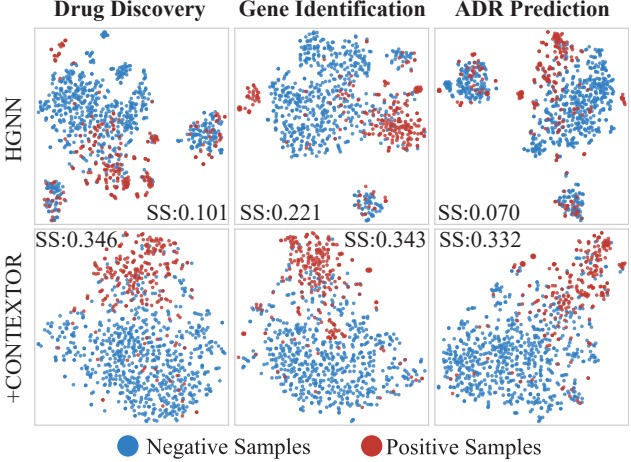

*Figure 5.* t-SNE visualizations of the learned representations under three scenarios. SS indicates the silhouette score, which quantifies the separability of clusters in the embedding space.

### 5.2.2. PERFORMANCE ON DRUG SYNERGY PREDICTION

Figure 3 summarizes the performance of our method and competing baselines on the Drug Synergy task across the NCI-ALMANAC and O'Neil datasets. Overall, our method consistently outperforms all baseline approaches under both random and stratified evaluation protocols.

On the (A) NCI-ALMANAC dataset, our method achieves the highest AUROC of 0.8580 under Random CV, exceeding HypergraphSynergy at 0.8530 and DeepSynergy at 0.8350, while also delivering superior AUPRC of 0.5664 and F1-score of 0.5323. Under the more challenging stratified evaluation on drug combinations, our method maintains a clear advantage with an AUROC of 0.7896, compared to 0.7798 for HypergraphSynergy and 0.7739 for DeepSynergy.

On the (B) O'Neil dataset, our method further demonstrates strong and robust performance. It achieves an AUROC of 0.9293 under Random CV, outperforming HypergraphSynergy at 0.9230 and DeepSynergy at 0.9060. Under stratified evaluations, our method attains AUROC values of 0.8655 for cell lines and 0.8738 for drug combinations, consistently surpassing HypergraphSynergy at 0.8554 and 0.8621, DeepSynergy at 0.8188 and 0.8583, and ComboFM at 0.8304 and 0.7555.

### 5.2.3. PERFORMANCE ON DGA DATASET

Figure 4 reports the performance of different models on the DGA dataset under three evaluation settings. Across all backbone architectures, including MLP, GCN, and HGNN, our method consistently outperforms the corresponding base

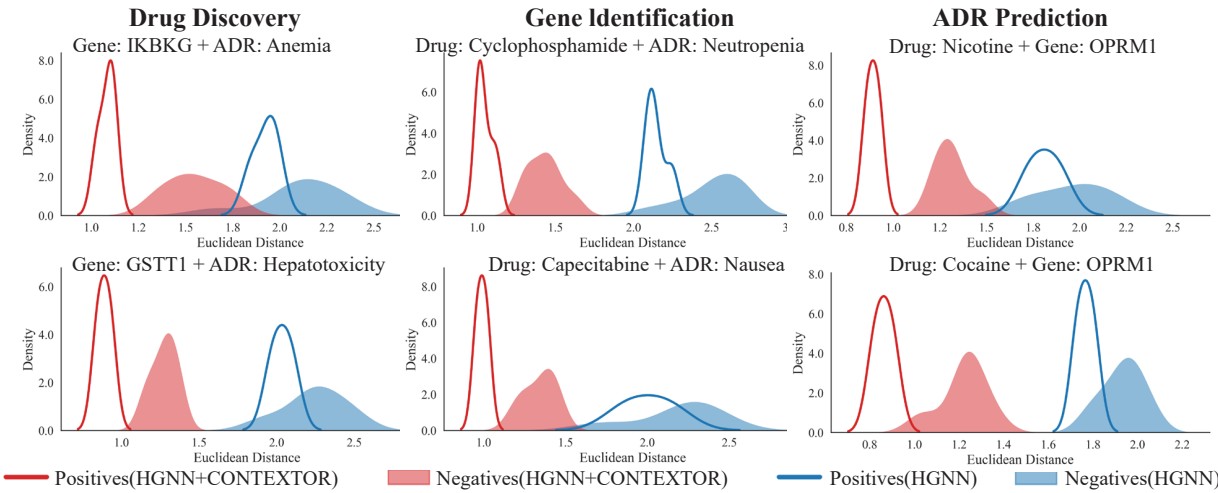

*Figure 6.* Case study on distance distributions of positive and negative high-order relations. CONTEXTOR substantially improves the discriminability between positive and negative relations.

models across all evaluation metrics. Under the Drug Discovery setting, MRR improves from 0.2481 to 0.2922 for MLP, from 0.2655 to 0.3315 for GCN, and from 0.2741 to 0.3180 for HGNN. Similar gains are observed in the Gene Identification setting, where MRR increases from 0.1854 to 0.2088 for MLP, from 0.1837 to 0.2277 for GCN, and from 0.2144 to 0.2466 for HGNN. The most substantial improvements occur in the ADR Prediction setting, with MRR rising from 0.3296 to 0.3867 for MLP, from 0.3444 to 0.4069 for GCN, and from 0.3684 to 0.4092 for HGNN. In addition to MRR, consistent improvements in AUROC and AUPRC are observed across all settings and architectures.

Detailed experimental results, parameter sensitivity, label efficiency and sparsity analysis are presented in *Supplementary Materials C.4, C.6, C.7 and C.8*, respectively.

### 5.3. Visualization of Feature Discriminability

The latent space is visualized using t-SNE (Maaten & Hinton, 2008) in Figure 5, illustrating the separation between positive and negative samples. Compared with HGNN, the embeddings learned with CONTEXTOR exhibit more compact intra-class structures and clearer inter-class separation, which is consistently reflected by higher Silhouette Scores. This suggests that incorporating contextual information helps mitigate the semantic entanglement induced by static representations, leading to a structured latent space that better aligns with the downstream prediction objective.

### 5.4. Case Study on Prediction Distributions

On the DGA dataset, for each task, we perform case studies on the most densely annotated query contexts, such as drug–ADR pairs linked to the largest number of genes in the gene identification setting. As shown in Figure 6, for drug

discovery, conditioning on the gene–ADR query contexts (i.e., IKBKG–Anemia and GSTT1–Hepatotoxicity) yields more compact distance distributions for positive drugs while clearly separating unrelated ones. Similarly, in gene identification and ADR prediction, HGNN+CONTEXTOR better distinguishes biologically plausible interactions from noisy alternatives. For example, under the Cyclophosphamide–Neutropenia context, genes involved in hematopoietic regulation are preferentially aligned, consistent with adverse reactions. Overall, the reduced overlap between positive and negative distance distributions suggests that conditioning on biological context enables the model to capture mechanism-specific high-order relations while suppressing irrelevant noise.

## 6. Conclusion

In this paper, we present CONTEXTOR, a general-purpose, plug-and-play contrastive learning framework for high-order relation inference that overcomes the limitations of static entity representation learning and symmetric relation inference in existing methods. By formulating high-order inference as a dynamic query–response process, CONTEXTOR conditions entity representations on relational context and supports asymmetric inference through an asymmetric conditional modulation mechanism. Extensive experiments across multiple biomedical benchmarks demonstrate consistent improvements across diverse backbone architectures, highlighting the effectiveness and generality of CONTEXTOR in data-sparse and combinatorial regimes. **Limitation.** While CONTEXTOR can in principle model higher-order relations of arbitrary order, our evaluations were performed on third-order relations due to the lack of large-scale higher-order benchmarks, and we leave broader evaluation in higher-order settings to future work.

## Acknowledgements

This work was supported by the National Key R&D Program of China (No.2025YFG0100700), the "Pioneer" and "Leading Goose" R&D Program of Zhejiang (No.2025C02068), the Natural Science Foundation of Zhejiang Province (No.LTGG24F020002, LY24F020013), the National Natural Science Foundation of China (No.62576304, 62102349), the Zhejiang Provincial Natural Science Foundation of China (No.LD26H300001), the Scientific Research Cultivation Fund of Hangzhou City University (No.J-202404), and the Scientific Research Cultivation Fund of Hangzhou City University: Special Project for Affiliated Hospitals (No.F-202408). The authors would like to acknowledge the Supercomputing Center of Hangzhou City University, for the support of advanced computing resources.

## Impact Statement

This paper presents work whose goal is to advance the field of machine learning. There are many potential societal consequences of our work, none of which we feel must be specifically highlighted here.

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

# Supplementary Material of "CONTEXTOR: Contextualized High-order Contrastive Learning"

## A. Notation

In this section, we summarize the commonly used concepts and notation in this paper for ease of reference, as shown in Table 2.

| Symbol | Description |
| --- | --- |
| $\mathcal{E}$ | The set of all entities in the dataset. |
| $e = (e_1, \ldots, e_m)$ | An ordered relational tuple of arity $m$. |
| $e_k$ | The $k$-th entity in a tuple, treated as the unobserved target. |
| $B$ | Batch size used during training. |
| $K$ | Total number of candidates (including negatives) in InfoNCE. |
| $\mathbf{h}_j \in \mathbb{R}^d$ | Base (static) embedding of entity $e_j$. |
| $\mathbf{z}_q \in \mathbb{R}^c$ | Query vector encoding the relational context. |
| $\gamma(\mathbf{z}_q), \beta(\mathbf{z}_q)$ | Query-dependent modulation parameters (scaling and shifting). |
| $\tilde{\mathbf{h}}_j$ (or $\mathbf{h}'_j$) | Contextualized embedding of the candidate entity. |
| $\mathcal{Q}(\cdot)$ | Query encoder function. |
| $\mathcal{M}(\cdot; \mathbf{z}_q)$ | Asymmetric Conditional Modulation (ACM) operator. |
| $s(\mathbf{z}_q, \mathbf{h}')$ | Query-response scoring function. |
| $\tau$ | Temperature hyperparameter in the contrastive loss. |

*Table 2.* Summary of Notations.

## B. Theoretical Analysis

In this section, we provide a theoretical analysis of the proposed CONTEXTOR framework. We first formally establish the inherent limitation of static embedding models in capturing semantic polysemy, and then demonstrate how the proposed ACM operator resolves this geometric bottleneck. We further analyze the gradient dynamics to show that CONTEXTOR enables context-aware optimization, and provide an information-theoretic interpretation of the contrastive learning objective by deriving a mutual information lower bound.

### B.1. Geometric Bottleneck of Static Embeddings

To rigorously characterize the challenge of "one entity, multiple meanings," we introduce the concept of the *Semantic*

*Polysemy Condition.*

**Definition B.1** (Semantic Polysemy Condition). Consider a polysemous entity $e \in \mathcal{E}$ involved in two distinct relational contexts triggered by queries $z_{q_1}$ and $z_{q_2}$. Let $t_1, t_2 \in \mathbb{R}^d$ be the ground-truth target representations required for these contexts. The entity $e$ is said to satisfy the *Semantic Polysemy Condition* if the optimal representations for the two contexts are significantly distinct, i.e., $\|t_1 - t_2\| > \epsilon$, for a sufficiently large threshold $\epsilon > 0$.

**Theorem B.2** (Impossibility of Static Resolution). *Let $f(\cdot)$ be a static embedding function such that $f(e) = \mathbf{h}_e \in \mathbb{R}^d$. Assuming a distance-based metric is minimized, a static model cannot arbitrarily approximate both targets $t_1$ and $t_2$ simultaneously with an error lower than $\epsilon/2$.*

*Proof Sketch.* By the triangle inequality in a metric space, for any static representation $\mathbf{h}_e$, the distances to the targets satisfy:

$$\|t_1 - t_2\| \leq \|t_1 - \mathbf{h}_e\| + \|\mathbf{h}_e - t_2\| \qquad (14)$$

Given the polysemy condition defined in Definition B.1 where $\|t_1 - t_2\| > \epsilon$, it follows that:

$$\|t_1 - \mathbf{h}_e\| + \|\mathbf{h}_e - t_2\| > \epsilon \qquad (15)$$

This inequality implies that the sum of alignment errors is lower-bounded by the semantic divergence of the contexts. Specifically, $\max(\|t_1 - \mathbf{h}_e\|, \|\mathbf{h}_e - t_2\|) > \epsilon/2$. Consequently, a static embedding is forced to converge to an average representation (the centroid), failing to capture the precise semantics of either context accurately. $\square$

### B.2. Resolution via Asymmetric Conditional Modulation

We now show that the proposed modulation mechanism provides sufficient expressivity to overcome the limitation established in Theorem B.2.

**Theorem B.3** (Expressivity of ACM). *Let the CONTEXTOR modulation operator be $\mathcal{M}(\mathbf{h}_e; z_q) = \gamma(z_q) \odot \mathbf{h}_e + \beta(z_q)$. For any pair of targets $t_1, t_2 \in \mathbb{R}^d$ and distinct queries $z_{q_1}, z_{q_2}$, there exist parameter sets for the modulation generators such that $\mathcal{M}(\mathbf{h}_e; z_{q_1}) = t_1$ and $\mathcal{M}(\mathbf{h}_e; z_{q_2}) = t_2$ simultaneously, provided the dimension $d$ is sufficient.*

*Proof Sketch.* The ACM operator introduces query-specific affine transformations. For context 1, we aim to solve $\gamma_1 \odot \mathbf{h}_e + \beta_1 = t_1$. For context 2, we solve $\gamma_2 \odot \mathbf{h}_e + \beta_2 = t_2$. Since $\gamma$ and $\beta$ are outputs of learnable functions dependent on $z_q$ (as defined in Eq. (8)), and $z_{q_1} \neq z_{q_2}$, the model can generate distinct transformation parameters $(\gamma_1, \beta_1)$ and $(\gamma_2, \beta_2)$. This effectively maps the single base point $\mathbf{h}_e$ to two arbitrarily distant regions in the latent space, thereby breaking the triangle inequality constraint of Theorem B.2 and resolving the semantic polysemy conflict. $\qquad\square$

### B.3. Context-Aware Gradient Dynamics

Finally, we analyze the backward pass to demonstrate how CONTEXTOR mitigates interference between conflicting contexts during optimization.

**Proposition B.4** (Gated Gradient Flow). *Consider the gradient of the loss $\mathcal{L}$ with respect to the base entity embedding $\mathbf{h}_j$. Applying the chain rule to the modulation equation, the update rule follows:*

$$\frac{\partial \mathcal{L}}{\partial \mathbf{h}_j} = \sum_k \frac{\partial \mathcal{L}}{\partial \mathbf{h}'_{j,k}} \cdot \frac{\partial \mathbf{h}'_{j,k}}{\partial \mathbf{h}_j} = \sum_k \frac{\partial \mathcal{L}}{\partial \mathbf{h}'_{j,k}} \odot \gamma(z_{q_k}) \quad (16)$$

*where $k$ indexes different queries in a mini-batch, and $\odot$ denotes the element-wise product.*

**Remark.** Unlike static models where gradients from all contexts are aggregated directly—often leading to destructive interference or catastrophic forgetting of rare contexts—CONTEXTOR scales the gradient by $\gamma(z_{q_k})$. This acts as a *soft gating mechanism*: if a feature dimension is irrelevant for the current query $z_{q_k}$ (i.e., $\gamma \approx 0$), the gradient flow is suppressed, thereby preserving the information in the base embedding for that dimension. This allows the model to learn context-specific features without overwriting shared knowledge.

### B.4. Information-Theoretic Derivation and Effective Tightness Analysis

In this subsection, we provide a formal information-theoretic analysis showing that the contrastive objective adopted in CONTEXTOR maximizes a variational lower bound on the mutual information between queries and query-conditioned entity representations. We further show that the proposed ACM mechanism improves the *effective tightness* of this bound by reducing the conditional entropy induced by semantic polysemy, as discussed in Section B.1.

B.4.1. VARIATIONAL LOWER BOUND OF INFONCE

Let $\mathbf{Z}_q$ denote the query representation and $\mathbf{H}'$ denote the query-conditioned entity embedding. The mutual informa-

tion between $\mathbf{Z}_q$ and $\mathbf{H}'$ is defined as

$$I(\mathbf{Z}_q; \mathbf{H}') = \mathbb{E}_{p(\mathbf{z}_q, \mathbf{h}')} \left[ \log \frac{p(\mathbf{h}' \mid \mathbf{z}_q)}{p(\mathbf{h}')} \right]. \quad (17)$$

Since the true density ratio is generally intractable, we employ the InfoNCE objective as a variational surrogate. Consider a mini-batch of size $K$ consisting of one positive pair $(\mathbf{z}_q, \mathbf{h}'_i)$ sampled from the joint distribution $p(\mathbf{z}_q, \mathbf{h}')$, and $K-1$ negative samples $\{\mathbf{h}'_j\}_{j \neq i}$ independently drawn from the marginal distribution $p(\mathbf{h}')$. Following (Oord et al., 2018), the InfoNCE objective is minimized when the scoring function approximates the density ratio, i.e.,

$$f^*(\mathbf{z}_q, \mathbf{h}') \propto \frac{p(\mathbf{h}' \mid \mathbf{z}_q)}{p(\mathbf{h}')}. \quad (18)$$

Substituting the optimal critic into the expected InfoNCE loss yields

$$
\begin{aligned}
-\mathcal{L}_{\text{InfoNCE}} &= \mathbb{E}\left[ \log \frac{\frac{p(\mathbf{h}'_i \mid \mathbf{z}_q)}{p(\mathbf{h}'_i)}}{\sum_{j=1}^{K} \frac{p(\mathbf{h}'_j \mid \mathbf{z}_q)}{p(\mathbf{h}'_j)}} \right] \\
&= \mathbb{E}\left[ \log \frac{p(\mathbf{h}'_i \mid \mathbf{z}_q)}{p(\mathbf{h}'_i)} \right. \\
&\qquad \left. - \log \left( \frac{p(\mathbf{h}'_i \mid \mathbf{z}_q)}{p(\mathbf{h}'_i)} + \sum_{j \neq i} \frac{p(\mathbf{h}'_j \mid \mathbf{z}_q)}{p(\mathbf{h}'_j)} \right) \right] \\
&= \mathbb{E}\left[ -\log \left( 1 + \frac{p(\mathbf{h}'_i)}{p(\mathbf{h}'_i \mid \mathbf{z}_q)} \sum_{j \neq i} \frac{p(\mathbf{h}'_j \mid \mathbf{z}_q)}{p(\mathbf{h}'_j)} \right) \right] \\
&\approx -\mathbb{E}\left[ \log \left( 1 + (K-1) \frac{p(\mathbf{h}'_i)}{p(\mathbf{h}'_i \mid \mathbf{z}_q)} \right) \right] \\
&\leq -\mathbb{E}\left[ \log \left( (K-1) \frac{p(\mathbf{h}'_i)}{p(\mathbf{h}'_i \mid \mathbf{z}_q)} \right) \right] \\
&= \mathbb{E}\left[ \log \frac{p(\mathbf{h}'_i \mid \mathbf{z}_q)}{p(\mathbf{h}'_i)} \right] - \log(K-1) \\
&= I(\mathbf{Z}_q; \mathbf{H}') - \log(K-1). \quad (19)
\end{aligned}
$$

Rearranging the above inequality yields the standard variational lower bound:

$$I(\mathbf{Z}_q; \mathbf{H}') \geq \log(K-1) - \mathcal{L}_{\text{InfoNCE}}. \quad (20)$$

B.4.2. IMPROVING THE EFFECTIVE TIGHTNESS VIA CONDITIONAL ENTROPY REDUCTION

While the lower bound in Eq. (20) holds for any encoder, its *effective tightness* depends on the maximum mutual information that the model can realize. Recalling the entropy decomposition of mutual information,

$$I(\mathbf{Z}_q; \mathbf{H}') = H(\mathbf{H}') - H(\mathbf{H}' \mid \mathbf{Z}_q), \quad (21)$$

maximizing mutual information is equivalent to minimizing the conditional entropy $H(\mathbf{H}' \mid \mathbf{Z}_q)$, i.e., reducing the uncertainty of the entity representation given the query.

**Failure of Static Models.** For static embedding models, where $\mathbf{H}' = \mathbf{H}_{\text{static}}$ is independent of $\mathbf{Z}_q$, the conditional entropy $H(\mathbf{H}_{\text{static}} \mid \mathbf{Z}_q)$ is lower-bounded by the inherent semantic polysemy of the entity. As established in Definition B.1 and Theorem B.2, a single static vector cannot simultaneously satisfy divergent geometric constraints induced by distinct queries. This limitation manifests as a diffuse and potentially multimodal conditional distribution $p(\mathbf{h} \mid \mathbf{z}_q)$, leading to high conditional entropy and a loose variational bound.

**Advantage of Asymmetric Conditional Modulation.** The ACM operator defined in Eq. (8) introduces a query-dependent transformation $\tilde{\mathbf{h}} = \mathcal{M}(\mathbf{h}; \mathbf{z}_q)$, which selectively emphasizes context-relevant semantics while suppressing irrelevant ones. As a result, the conditional distribution of the query-conditioned representation becomes more concentrated and better aligned with the query context. This suggests that, in expectation,

$$H(\tilde{\mathbf{H}}_{\text{ACM}} \mid \mathbf{Z}_q) \;\lesssim\; H(\mathbf{H}_{\text{static}} \mid \mathbf{Z}_q). \tag{22}$$

Importantly, ACM does not alter the analytical form of the InfoNCE lower bound in Eq. (20). Instead, by reducing the conditional entropy induced by semantic polysemy, it increases the attainable mutual information $I(\mathbf{Z}_q; \mathbf{H}')$. Consequently, for a fixed batch size $K$, the InfoNCE objective can more closely approximate the true mutual information, yielding a practically tighter bound compared to static embedding models. This analysis highlights that asymmetric conditional modulation is not merely an architectural choice, but a necessary mechanism for effective contrastive learning in polysemous relational settings.

## C. Supplementary Experiments

### C.1. Dataset Detail

We conduct experiments on three biologically meaningful entity association datasets: DMD (Drug–Microbe–Disease), Drug Synergy, and DGA (Drug–Gene–ADR). The dataset statistics are summarized in Table 3. Detailed descriptions of the datasets are provided below.

| Dataset | #Entity 1 | #Entity 2 | #Entity 3 | #Triplets |
|---------|-----------|-----------|-----------|-----------|
| DMD | 270 | 58 | 167 | 2,763 |
| ALMANAC | 87 | 87 | 55 | 74,139 |
| O'Neil | 38 | 38 | 39 | 18,950 |
| DGA | 217 | 384 | 180 | 1,270 |

*Table 3.* Statistics of the datasets used for evaluation.

#### C.1.1. DMD DATASET

The DMD dataset consists of 2,763 drug–microbe–disease triplets constructed by integrating drug–microbe and microbe–disease associations from MDAD (Sun et al., 2018) and HMDAD (Ma et al., 2017). It involves 270 drugs, 58 microbes, and 167 diseases. Drug representations are derived from SMILES strings obtained from PubChem (Kim et al., 2019), while microbe and disease features are based on NCBI Taxonomy and MeSH descriptors, respectively.

#### C.1.2. DRUG SYNERGY DATASET

The drug synergy dataset is collected from two large-scale high-throughput screening resources: NCI-ALMANAC (Holbeck et al., 2017) and O'Neil (O'Neil et al., 2016). It contains drug–drug–cell-line triplets annotated with synergy scores. Drugs are represented using SMILES strings from PubChem, and cell lines are characterized by log-transformed and z-score normalized gene expression profiles of 651 cancer-related genes. After removing incomplete samples, the final datasets include 74,139 triplets (87 drugs, 55 cell lines) from NCI-ALMANAC and 18,950 triplets (38 drugs, 39 cell lines) from O'Neil.

#### C.1.3. DGA DATASET

We construct a new Drug–Gene–ADR (DGA) dataset by integrating curated annotations from PharmGKB (Gong et al., 2021) and DrugBank (Knox et al., 2024). All adverse drug reaction (ADR) terms are standardized using UMLS Concept Unique Identifiers (CUIs) (Bodenreider, 2004) to ensure consistency across sources. The resulting dataset contains 1,270 manually curated triplets, covering 217 drugs, 384 genes, and 180 ADRs.

### C.2. Baseline Models and Experimental Settings

This section describes the task formulation, baseline models, and experimental configurations for each dataset. For fair comparison, all baseline implementations and hyperparameter settings are adopted directly from the corresponding original studies.

#### C.2.1. DMD DATASET

The task on the DMD dataset is to predict drug–microbe–disease ternary associations based on known triplets. This problem involves high-order relational reasoning across heterogeneous biological entities.

We compare our method with a wide range of baseline models, including RF, MLP, CP, Tucker, and CoSTCo; graph-based neural networks such as GCN (Kipf, 2016), GraphSAGE, GAT, and GIN; as well as hypergraph-based models including DHNE (Tu et al., 2018), Hyper-SAGNN (Zhang et al., 2019), HGSynergy (Liu et al., 2022),

| Task | Method | ALMANAC | | | O'Neil | | |
|---|---|---|---|---|---|---|---|
| | | AUROC | AUPRC | F1-Score | AUROC | AUPRC | F1-Score |
| Random level | DeepSynergy | 0.8350 | 0.4964 | 0.4880 | 0.9060 | 0.5638 | 0.5445 |
| | ComboFM | 0.8180 | 0.4887 | 0.4753 | 0.8945 | 0.5362 | 0.4655 |
| | DTF | 0.8238 | 0.4752 | 0.4688 | 0.9138 | 0.5829 | 0.5828 |
| | Celebi's method | 0.7919 | 0.4533 | 0.4489 | 0.8783 | 0.5338 | 0.5249 |
| | Li's method | 0.7917 | 0.4246 | 0.4404 | 0.8750 | 0.5141 | 0.5181 |
| | Hypergraph Synergy | 0.8530 | 0.5572 | 0.5295 | 0.9230 | 0.6328 | 0.6025 |
| | **CONTEXTOR** | **0.8581** | **0.5664** | **0.5323** | **0.9293** | **0.6419** | **0.6126** |
| Cell line level | DeepSynergy | 0.7035 | 0.2772 | 0.3371 | 0.8188 | 0.3455 | 0.4015 |
| | ComboFM | 0.7477 | 0.4205 | 0.4017 | 0.8304 | 0.3620 | 0.2431 |
| | DTF | 0.7551 | 0.3949 | 0.4132 | 0.8387 | 0.4022 | 0.4371 |
| | Celebi's method | 0.7603 | 0.4128 | 0.4164 | 0.8322 | 0.3905 | 0.4181 |
| | Li's method | 0.7739 | 0.4474 | 0.4409 | 0.8371 | 0.4380 | 0.4484 |
| | Hypergraph Synergy | **0.7886** | 0.4505 | 0.4455 | 0.8554 | 0.4610 | 0.4627 |
| | **CONTEXTOR** | 0.7884 | **0.4534** | **0.4475** | **0.8655** | **0.4706** | **0.4747** |
| Drug pairs level | DeepSynergy | 0.7739 | 0.3626 | 0.4104 | 0.8583 | 0.4555 | 0.4723 |
| | ComboFM | 0.7555 | 0.3787 | 0.3667 | 0.7555 | 0.4387 | 0.3892 |
| | DTF | 0.7502 | 0.3165 | 0.3613 | 0.8424 | 0.3985 | 0.4307 |
| | Celebi's method | 0.7369 | 0.3149 | 0.3704 | 0.7897 | 0.3386 | 0.3624 |
| | Li's method | 0.7384 | 0.3270 | 0.3632 | 0.7764 | 0.3182 | 0.3593 |
| | Hypergraph Synergy | 0.7798 | 0.3795 | 0.4128 | 0.8621 | 0.4811 | 0.4838 |
| | **CONTEXTOR** | **0.7896** | **0.3838** | **0.4154** | **0.8738** | **0.5001** | **0.5064** |

*Table 4.* Detailed quantitative results of model performance on the Drug Synergy task under Random and Stratified cross-validation settings for the ALMANAC and O'Neil datasets.

| Model | Drug Discovery | | | Gene Identification | | | ADR Prediction | | |
|---|---|---|---|---|---|---|---|---|---|
| | AUROC (%) | AUPRC (%) | MRR | AUROC (%) | AUPRC (%) | MRR | AUROC (%) | AUPRC (%) | MRR |
| MLP | $88.97_{\pm1.42}$ | $74.96_{\pm4.21}$ | $0.2481_{\pm0.0393}$ | $84.29_{\pm0.63}$ | $70.74_{\pm3.58}$ | $0.1854_{\pm0.0305}$ | $90.78_{\pm1.38}$ | $80.03_{\pm2.35}$ | $0.3296_{\pm0.0141}$ |
| **+CONTEXTOR** | $\mathbf{91.87}_{\pm1.19}$ | $\mathbf{82.39}_{\pm2.25}$ | $\mathbf{0.2922}_{\pm0.0309}$ | $\mathbf{87.46}_{\pm1.22}$ | $\mathbf{76.73}_{\pm2.98}$ | $\mathbf{0.2088}_{\pm0.0217}$ | $\mathbf{92.84}_{\pm0.39}$ | $\mathbf{83.97}_{\pm1.71}$ | $\mathbf{0.3867}_{\pm0.0265}$ |
| GCN | $88.34_{\pm2.41}$ | $74.95_{\pm5.03}$ | $0.2655_{\pm0.0330}$ | $83.03_{\pm0.80}$ | $71.06_{\pm0.92}$ | $0.1837_{\pm0.0226}$ | $89.25_{\pm1.85}$ | $77.07_{\pm4.62}$ | $0.3444_{\pm0.0423}$ |
| **+CONTEXTOR** | $\mathbf{92.15}_{\pm1.17}$ | $\mathbf{83.99}_{\pm2.54}$ | $\mathbf{0.3315}_{\pm0.0438}$ | $\mathbf{85.91}_{\pm1.44}$ | $\mathbf{78.30}_{\pm1.71}$ | $\mathbf{0.2277}_{\pm0.0319}$ | $\mathbf{91.20}_{\pm1.15}$ | $\mathbf{84.54}_{\pm0.61}$ | $\mathbf{0.4069}_{\pm0.0383}$ |
| HGNN | $84.50_{\pm3.08}$ | $69.58_{\pm10.61}$ | $0.2741_{\pm0.0352}$ | $85.64_{\pm3.37}$ | $72.45_{\pm4.18}$ | $0.2144_{\pm0.0237}$ | $85.46_{\pm2.23}$ | $69.28_{\pm8.02}$ | $0.3684_{\pm0.0177}$ |
| **+CONTEXTOR** | $\mathbf{91.57}_{\pm2.40}$ | $\mathbf{82.04}_{\pm5.71}$ | $\mathbf{0.3180}_{\pm0.0372}$ | $\mathbf{91.61}_{\pm1.55}$ | $\mathbf{84.57}_{\pm2.15}$ | $\mathbf{0.2466}_{\pm0.0118}$ | $\mathbf{92.94}_{\pm1.03}$ | $\mathbf{84.61}_{\pm2.45}$ | $\mathbf{0.4092}_{\pm0.0402}$ |

*Table 5.* Performance on the DGA dataset under three different evaluation settings. Results are reported as mean $\pm$ standard deviation.

and MCHNN (Liu et al., 2023).

These baselines are selected to cover different modeling paradigms, ranging from non-graph tensor factorization methods to graph neural networks and hypergraph-based approaches that explicitly capture high-order interactions, which are particularly relevant for ternary association prediction.

All experimental settings, including data splits, model architectures, optimization strategies, and hyperparameters, are directly adopted from MCHNN (Liu et al., 2023). No additional tuning is performed for the baseline models.

### C.2.2. DRUG SYNERGY DATASET

The task on the drug synergy dataset is to predict the synergy score of drug pairs under specific cell-line contexts, formulated as a drug–drug–cell-line triplet prediction problem.

The baseline methods include DeepSynergy (Preuer et al., 2018), DTF (Sun et al., 2020), ComboFM (Julkunen et al., 2020), Celebi's method (Celebi et al., 2019), Li's method (Li

et al., 2018), and HGSynergy (Liu et al., 2022).

These methods are chosen as they represent commonly used deep learning, factorization-based, and network-based approaches for drug synergy prediction, and have been shown to achieve competitive performance on this dataset in prior studies.

We follow the experimental configuration reported in HGSynergy (Liu et al., 2022), including model architectures, training protocols, and hyperparameter settings. All baseline results are obtained using the original implementations without modification.

### C.2.3. DGA DATASET

The DGA dataset focuses on predicting drug–gene–adverse drug reaction (ADR) associations, which requires modeling high-order interactions among pharmacological, genomic, and clinical entities.

We evaluate our method under three representative backbone models: MLP, GCN, and HGNN (Feng et al., 2019).

These backbones are selected to assess the effectiveness of the proposed method across different structural assumptions, including independent triplet modeling (MLP), pairwise message passing on graphs (GCN), and explicit high-order relation modeling via hypergraphs (HGNN).

For each positive triplet $(d, g, a)$, negative samples are generated by independently corrupting the drug, gene, or ADR. All backbone models are implemented using the original architectures and training settings described in their respective studies, without additional tuning.

### C.2.4. DATASET-SPECIFIC SETTINGS

**DMD Dataset.** Following (Liu et al., 2023), we employ a 5-fold cross-validation protocol. Evaluation includes four negative sampling strategies: (i) *drug-level*, (ii) *microbe-level*, and (iii) *disease-level*, where the respective entity is replaced with a random one from the training set; and (iv) *random negative sampling*, where an arbitrary entity in the triplet is replaced. The model is trained for 3,000 epochs using Adam (Kingma & Ba, 2014) (lr=$5 \times 10^{-3}$, weight decay=$1 \times 10^{-4}$) with early stopping based on validation loss. We replace the baseline's DGI module with our CONTEXTOR method ($\tau = 0.07$), setting the auxiliary contrastive loss weight to $\lambda = 0.2$.

**Drug Synergy Dataset.** We adopt the backbone architecture from HypergraphSynergy (Liu et al., 2022), comprising a GCN for drugs ($L = 3$), an FCN for cell lines ($L = 3$), and an HGNN for high-order relations ($L = 2$). The original reconstruction loss is replaced by CONTEXTOR ($\tau = 0.07$), with the balance hyperparameter fixed at $\lambda = 0.4$. The model is optimized via Adam (Kingma & Ba, 2014) (lr=$1 \times 10^{-4}$).

**DGA Dataset.** Experiments are conducted under a 5-fold CV protocol, reporting results averaged across folds. For each fold, the best checkpoint is selected based on validation MRR. Models are optimized using Adam (Kingma & Ba, 2014) (lr=$1 \times 10^{-4}$, weight decay=$1 \times 10^{-4}$), with the auxiliary contrastive loss weight set to $\lambda = 0.9$.

### C.3. Evaluation Metrics and Computation Details

We evaluate model performance using both classification-based and ranking-based metrics, including AUROC, AUPRC, F1-score, Hit Ratio (Hits@K), NDCG@K, and Mean Reciprocal Rank (MRR).

**Classification Metrics.** Given predicted scores $s_i \in \mathbb{R}$ and binary labels $y_i \in \{0, 1\}$ for test triplets, AUROC measures the probability that a randomly chosen positive instance is ranked higher than a randomly chosen negative instance, while AUPRC summarizes the precision–recall trade-off across different decision thresholds.

The F1-score is defined as:

$$\text{F1} = \frac{2 \cdot \text{Precision} \cdot \text{Recall}}{\text{Precision} + \text{Recall}}, \qquad (23)$$

where

$$\text{Precision} = \frac{\text{TP}}{\text{TP} + \text{FP}}, \quad \text{Recall} = \frac{\text{TP}}{\text{TP} + \text{FN}}. \quad (24)$$

**Ranking Metrics.** For ranking-based evaluation, each positive test triplet is evaluated against a set of candidate entities constructed according to the task-specific prediction setting.

The Hit Ratio at $K$ (Hits@K) is computed as:

$$\text{Hits@K} = \frac{1}{|\mathcal{Q}|} \sum_{q \in \mathcal{Q}} \mathbb{I}(\text{rank}_q \leq K), \qquad (25)$$

where $\mathcal{Q}$ denotes the set of test queries, $\text{rank}_q$ is the rank position of the ground-truth entity for query $q$, and $\mathbb{I}(\cdot)$ is the indicator function.

The Normalized Discounted Cumulative Gain at $K$ (NDCG@K) is defined as:

$$\text{NDCG@K} = \frac{1}{|\mathcal{Q}|} \sum_{q \in \mathcal{Q}} \frac{\text{DCG@K}_q}{\text{IDCG@K}_q}, \qquad (26)$$

where

$$\text{DCG@K}_q = \sum_{i=1}^{K} \frac{2^{\text{rel}_{q,i}} - 1}{\log_2(i + 1)}, \qquad (27)$$

and $\text{rel}_{q,i} \in \{0, 1\}$ indicates whether the entity ranked at position $i$ corresponds to the ground-truth target.

**Mean Reciprocal Rank (MRR).** The Mean Reciprocal Rank is computed as:

$$\text{MRR} = \frac{1}{|\mathcal{Q}|} \sum_{q \in \mathcal{Q}} \frac{1}{\text{rank}_q}. \qquad (28)$$

On the DGA dataset, we adopt task-specific candidate construction strategies consistent with the negative sampling process. For the Drug Discovery task, we fix the (gene, ADR) pair and rank all candidate drugs by their predicted scores. For the Gene Identification and ADR Prediction tasks, we respectively fix the (drug, ADR) and (drug, gene) pairs and rank all candidate genes or ADRs. For each positive test triplet, the rank of the ground-truth entity is obtained by sorting all candidates in descending order of predicted scores. The reciprocal rank is computed for each positive instance, and MRR is reported as the average over all test positives.

| Model | Drug Discovery | | | Gene Identification | | | ADR Prediction | | |
|---|---|---|---|---|---|---|---|---|---|
| | AUROC (%) | AUPRC (%) | MRR | AUROC (%) | AUPRC (%) | MRR | AUROC (%) | AUPRC (%) | MRR |
| w/o ACM | 89.68 ± 1.75 | 77.83 ± 6.64 | 0.2853 ± 0.0350 | 89.25 ± 1.57 | 77.96 ± 1.76 | 0.2270 ± 0.0359 | 91.13 ± 1.59 | 79.75 ± 4.30 | 0.3845 ± 0.0227 |
| **CONTEXTOR** | **91.57 ± 2.40** | **82.04 ± 5.71** | **0.3180 ± 0.0372** | **91.61 ± 1.55** | **84.57 ± 2.15** | **0.2466 ± 0.0118** | **92.94 ± 1.03** | **84.61 ± 2.45** | **0.4092 ± 0.0402** |

*Table 6.* Ablation study on the DGA dataset under three different evaluation settings. Results are reported as mean ± standard deviation.

## C.4. Detailed Experimental Results

Table 4 summarizes the model performance on the Drug Synergy task evaluated using random and stratified cross-validation on the ALMANAC and O'Neil datasets. We report the results in Table 5, which summarizes the performance on the DGA dataset under three different evaluation settings. Results are reported as mean ± standard deviation.

## C.5. Ablation Study

To quantitatively assess the contribution of the asymmetric modulation mechanism, we conduct an ablation study on the DGA dataset. This modulation mechanism is a core component of our Drug-Gene-ADR framework, designed to disentangle condition-specific embeddings by dynamically modulating candidate entity representations based on the specific query context.

In this ablation analysis, we introduce a variant denoted as **w/o ACM**. For this variant, we replace the ACM module with a static linear projection. To ensure a strictly fair comparison, this static projection is designed to maintain the exact same parameter scale as the original modulation module.

As summarized in Table 6, removing the asymmetric modulation results in a consistent and significant performance degradation across all evaluation metrics in the Drug Discovery, Gene Identification, and ADR Prediction tasks. For instance, in the Drug Discovery setting, the MRR strictly decreases from 0.3180 to 0.2853, alongside notable drops in both AUROC and AUPRC. These results empirically validate our theoretical premise: static embeddings inherently suffer from semantic entanglement when forced to model complex, context-dependent biomedical relations. Ultimately, the query-aware modulation is essential for dynamically adapting entity semantics and enabling accurate Drug-Gene-ADR relation inference.

## C.6. Parameter Sensitivity Analysis

We conducted comprehensive sensitivity experiments on the DGA dataset across three distinct scenarios: drug discovery, gene identification, and ADR prediction. Our analysis focused on determining the optimal values for two critical hyperparameters: the contrastive loss weight $\alpha$ and the depth of ACM layers $L$. The parameter search space was defined as $\alpha \in \{0.0, 0.1, \ldots, 0.9\}$ and $L \in \{1, 2, 3, 4\}$. As

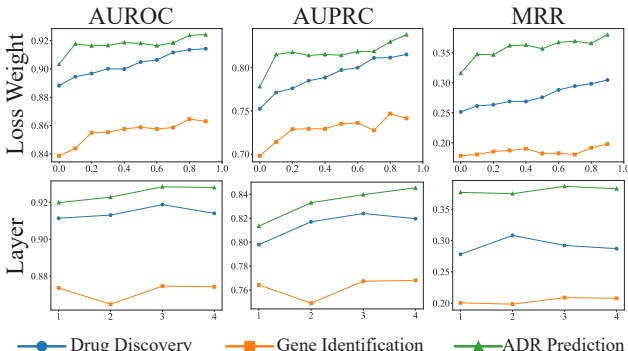

*Figure 7.* Hyperparameter study for CONTEXTOR.

shown in Figure 7, across all three tasks, we observed a consistent positive correlation between the contrastive weight and model performance, validating the efficacy of the auxiliary supervision. Consequently, we selected $\alpha = 0.9$ for the final model configuration to maximize the utilization of contextual signals. Regarding the model depth, while CONTEXTOR demonstrated robustness across varying layers, it achieved the best global performance at $L = 3$. This setting was therefore adopted to effectively capture high-order structural dependencies without succumbing to over-smoothing.

## C.7. Label Efficiency Analysis

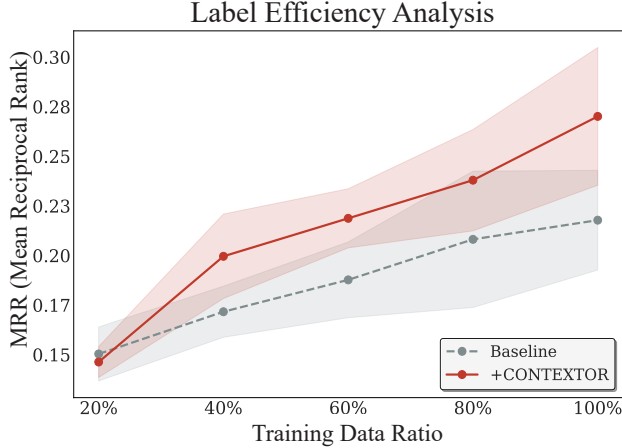

*Figure 8.* Label Efficiency Analysis. Comparison of MRR between the Baseline (dashed line) and +CONTEXTOR (solid line) across varying training data ratios (from 20% to 100%).

We further investigate the capability of +CONTEXTOR to

learn from limited supervision by varying the training data ratio from 20% to 100%. As illustrated in Figure 8, our proposed method demonstrates significantly superior label efficiency compared to the baseline. While both models benefit from increased data availability, +CONTEXTOR exhibits a much steeper performance trajectory. Notably, our method achieves an MRR of approximately 0.20 with only 40% of the training data, effectively surpassing the performance of the baseline model trained on 60% of the data. This gap widens as the dataset grows, culminating in a substantial performance lead at the 100% ratio. These results provide strong empirical evidence that +CONTEXTOR is highly effective at extracting rich semantic signals from sparse labels, making it a robust solution for data-constrained scenarios.

## C.8. Sparsity Analysis

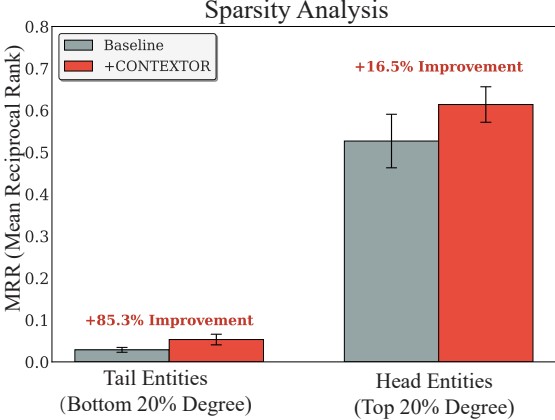

*Figure 9.* Sparsity Analysis. Comparison of MRR performance on Tail Entities (bottom 20% degree) and Head Entities (top 20% degree).

To investigate the model's robustness against structural sparsity, we categorize test nodes into two groups based on their connectivity: "Tail Entities" (bottom 20% degree) and "Head Entities" (top 20% degree). As shown in Figure 9, while +CONTEXTOR achieves a solid 16.5% improvement on the information-rich head entities, the gains are significantly more pronounced on the sparse tail entities, reaching an 85.3% relative improvement over the baseline. This disparity in performance gain highlights a critical advantage of our approach: by integrating contextual semantics, +CONTEXTOR effectively compensates for the lack of topological information in low-degree nodes, thereby resolving the long-tail issue prevalent in standard graph learning models.

## C.9. Generalization on Non-Biomedical Benchmarks

Although the primary motivation of our framework stems from the prevalence and scientific significance of high-order relational structures in biomedical discoveries, CON-

| Dataset | Model | AUROC (%) | AUPRC (%) |
|---------|-------|-----------|-----------|
| Elliptic++ | HGNN | 91.13 ± 5.15 | 82.13 ± 5.66 |
| | +CONTEXTOR | **94.08 ± 2.41** | **90.25 ± 3.11** |
| DBLP | HGNN | 78.36 ± 12.36 | 54.87 ± 22.53 |
| | +CONTEXTOR | **83.29 ± 5.03** | **60.69 ± 12.95** |

*Table 7.* Performance evaluation on non-biomedical hypergraph datasets (Elliptic++ and DBLP) under the link prediction setting. Results are reported as mean ± standard deviation.

TEXTOR is fundamentally designed as a general-purpose methodology. To demonstrate that the proposed framework is domain-agnostic and generalizes well to standard non-biomedical hypergraph datasets, especially where semantic signals might be weak, noisy, or structurally different, we conduct additional experiments on two widely used benchmarks: Elliptic++ (financial transaction networks) (El-mougy & Liu, 2023) and DBLP (academic citation and co-authorship networks) (Kong et al., 2012).

For the experimental setup, we evaluate the models under a standard high-order link prediction setting. We utilize all 4,339 available triplets for the DBLP dataset and uniformly sample 2,000 triplets for the Elliptic++ dataset. We compare the base Hypergraph Neural Network (HGNN) with its variant augmented by our plug-and-play CONTEXTOR module.

As presented in Table 7, the integration of CONTEXTOR consistently and substantially improves performance across both non-biomedical datasets. On the Elliptic++ dataset, CONTEXTOR boosts the AUPRC from 82.13% to 90.25%. Similarly, on the DBLP dataset, it yields a notable gain in AUROC from 78.36% to 83.29%.

Furthermore, it is worth highlighting that the integration of CONTEXTOR significantly reduces the standard deviation across all metrics (e.g., the standard deviation of AUPRC on DBLP drops from ±22.53 to ±12.95). This indicates that our query-driven contrastive alignment not only improves absolute predictive performance but also drastically enhances the stability and robustness of the representation learning process. These results confirm that our approach effectively generalizes well beyond biomedical data and remains highly competitive in diverse structural scenarios.

## C.10. Integration with Semantic-Aware Models

While some advanced hypergraph models, such as Natural-HNN and HSDN, are specifically designed to automatically disentangle hyperedge semantics, we investigate whether the proposed framework provides additional synergistic value when integrated with these architectures.

To explore this, we integrated the proposed framework with Natural-HNN (Lee et al., 2026), a state-of-the-art model

| Model | Drug Discovery | | | Gene Identification | | | ADR Prediction | | |
|---|---|---|---|---|---|---|---|---|---|
| | AUROC (%) | AUPRC (%) | MRR | AUROC (%) | AUPRC (%) | MRR | AUROC (%) | AUPRC (%) | MRR |
| Natural-HNN | $90.15 \pm 1.41$ | $78.28 \pm 3.12$ | $0.3179 \pm 0.0312$ | $89.65 \pm 0.57$ | $80.94 \pm 1.18$ | $0.2121 \pm 0.0216$ | $90.54 \pm 0.83$ | $77.54 \pm 1.43$ | $0.4082 \pm 0.0407$ |
| **+CONTEXTOR** | **$94.59 \pm 0.68$** | **$89.15 \pm 0.83$** | **$0.3936 \pm 0.0235$** | **$91.70 \pm 2.01$** | **$84.62 \pm 3.90$** | **$0.2749 \pm 0.0218$** | **$95.29 \pm 0.54$** | **$91.22 \pm 0.71$** | **$0.5129 \pm 0.0248$** |

*Table 8.* Performance comparison on the DGA dataset integrating the CONTEXTOR with semantic-aware models. Results are reported as mean $\pm$ standard deviation.

that disentangles hyperedges through the lens of category theory. As summarized in Table 8, the combination yields remarkable synergistic improvements across all evaluation tasks. Most notably, in the ADR Prediction task, the integration boosts Natural-HNN's MRR from 0.4082 to 0.5129. These results confirm that our contrastive alignment strategy is orthogonal and highly complementary to models that inherently perform semantic disentanglement.

## C.11. Comparison with Large Language Models

To further evaluate the practical competitiveness of CON-TEXTOR against modern foundation models, we conduct additional experiments using GPT-5.2 and Gemini-3.1 under the same DGA evaluation protocol. Specifically, the LLMs are evaluated in a zero-shot ranking setting, where each model is provided with an incomplete biomedical triplet and asked to rank candidate entities according to their likelihood of completing a biologically plausible (Drug, Gene, ADR) relation.

For fair comparison, all LLMs use a unified prompt template and deterministic decoding with temperature set to 0. Candidate ordering is randomized across evaluation runs to reduce positional bias. Furthermore, models are constrained to return only a ranked JSON list without additional explanations or formatting.

| Model | Drug Discovery | Gene Identification | ADR Prediction |
|---|---|---|---|
| Gemini-3.1 | $0.3449 \pm 0.0293$ | $0.2364 \pm 0.0194$ | $0.3478 \pm 0.0532$ |
| GPT-5.2 | **$0.3596 \pm 0.0456$** | $0.2314 \pm 0.0163$ | $0.3682 \pm 0.0485$ |
| **CONTEXTOR** | $0.3180 \pm 0.0372$ | **$0.2466 \pm 0.0118$** | **$0.4092 \pm 0.0402$** |

*Table 9.* MRR comparison on the DGA dataset between CONTEX-TOR and recent foundation models. Results are reported as mean $\pm$ standard deviation.

As summarized in Table 9, CONTEXTOR achieves the best performance on the Gene Identification and ADR Prediction tasks, while GPT-5.2 performs slightly better on Drug Discovery. One possible explanation is that drug-related associations are more extensively represented in large-scale biomedical corpora, potentially benefiting pretrained language models through richer parametric knowledge.

In contrast, Gene Identification and ADR Prediction require modeling finer-grained conditional dependencies among biomedical entities and stronger generalization to less-studied relations. The strong performance of CONTEXTOR

on these tasks suggests that explicit structured relational modeling can provide complementary advantages beyond purely text-based representations.

In addition, CONTEXTOR operates without reliance on external API-based inference, making it computationally efficient and potentially practical for controlled biomedical research environments.

To ensure a reproducible and rigorous evaluation, we employ a highly structured zero-shot ranking prompt. The prompt is designed to align with our asymmetric query-response paradigm by forcing the LLMs to rank a fixed set of candidates based on an incomplete triplet context. The general prompt template is constructed as follows:

**System Instruction:** You are an expert in clinical pharmacology, pharmacogenomics, and biological network analysis. Your task is to perform high-order relational reasoning to identify valid combinations of Drugs, Genes, and Adverse Drug Reactions (ADRs).

**Task Description:** You are given an incomplete biomedical triplet acting as the context. Your objective is to rank a provided list of candidate entities based on their biochemical and pharmacological likelihood to complete a valid (Drug, Gene, ADR) triplet.

**Input Context:** *[Dynamically injected based on the evaluation scenario]*

- *ADR Prediction:* Context: The patient is administered the drug [Drug Name], which interacts with the gene [Gene Name]. Target Entity Type: Adverse Drug Reaction (ADR).
- *Gene Identification:* Context: The patient is administered the drug [Drug Name] and subsequently develops the adverse drug reaction [ADR Name]. Target Entity Type: Gene.
- *Drug Discovery:* Context: We are investigating the genetic mechanism where the gene [Gene Name] is associated with the adverse drug reaction [ADR Name]. Target Entity Type: Drug.

**Candidate List:** Please rank the following candidate entities from most likely to least likely to complete the triplet context described above.

**Output Formatting Constraints:** You must output your prediction strictly as a ranked JSON list of the candidate names, without any additional text, explanation, or markdown formatting.
Example format: `["Candidate_B", "Candidate_A", "Candidate_C", ...]`

