# OpenReview forum: "CONTEXTOR: Contextualized High-order Contrastive Learning"
_ICML.cc/2026/Conference — ICML 2026 regular_

### Official Review · Reviewer_ryY6 · 2026-02-25

**Soundness:** 2
**Presentation:** 3
**Significance:** 2
**Originality:** 2
**Overall Recommendation:** 3
**Confidence:** 3

**Summary:**

This paper addresses high-order relational learning in biomedical domains, where existing methods rely on static entity representations and symmetric relation assumptions that fail to capture contextual and asymmetric interactions. The authors propose CONTEXTOR, a plug-and-play framework that models high-order relation inference as a dynamic query–response process by decomposing relations into incomplete query tuples and response entities, and applying asymmetric conditional modulation with multi-fold contrastive learning.

**Compliance With Llm Reviewing Policy:**

Affirmed.

**Final Justification:**

My concern has been basically resolved, but I still have a certain degree of disagreement with the meaing of using graph and contrastive learning.

**Key Questions For Authors:**

1. Why totally ingnore LLM-based methods which are demonstrated effective for biomedical extraction tasks?

**Limitations:**

I did not see any limitations discussion.

**Strengths And Weaknesses:**

Strengths
1.  paper focuses on the task of Drug–Gene–ADR extraction, which is both meaningful and clinically important. By addressing this complex high-order relation, the study targets a problem with clear biomedical relevance and practical impact.

2. The paper is structured in a clear and coherent manner, making it easy to comprehend.

3. The authors provide publicly available code and detailed implementation details, enhancing the study’s transparency and reproducibility.

Weaknesses
1. The paper claims that “most existing approaches rely on static entity representations, implicitly assuming fixed functions or semantics.” I disagree with this characterization. In recent years, many approaches—particularly LLM-based and contextualized embedding methods—have explicitly modeled context-dependent entity representations. Therefore, the claim that most existing approaches use static embeddings appears overstated and requires more careful justification and updated literature support.

2. The manuscript also states that “existing contrastive learning methods predominantly focus on pairwise relations.” This statement seems insufficiently supported. Asymmetric and high-order relation extraction has been studied for several years, and there exist prior works that explicitly address non-symmetric and multi-entity relational modeling. The authors should clarify how their formulation differs substantively from these existing methods.

3. The selected baselines are relatively outdated, with several proposed in 2016–2018 and others in 2022–2023. Although the paper focuses on contrastive learning, completely excluding comparisons with modern LLM-based methods is difficult to justify. At a minimum, the authors should compare against strong contemporary LLM-based baselines (e.g., GPT-5, Gemini 3, or other biomedical-adapted LLMs), or provide a clear explanation for why such comparisons are not included. Without these comparisons, it is difficult to assess the practical competitiveness and relevance of the proposed method.

---

> ### Author Rebuttal · Authors · 2026-03-31
>
> We sincerely thank you for your insightful feedback and recognition of our work. We remain fully open to implementing any additional revisions. Below, we address each of your comments in detail.
>
> >**Q1. Scope of "Static" Entity Representations:** The paper claims that “most existing approaches rely on static entity representations.” I disagree with this characterization, as many recent LLM-based and contextualized methods explicitly model context-dependent representations. This claim appears overstated and requires clarification.
>
> **A1:** We fully agree that modern LLMs/PLMs utilize contextualized representations. Our claim, however, specifically targets the downstream stage of **structured graph/hypergraph-based high-order relation inference** (e.g., predicting drug-gene-ADR tuples).
>
> In this specific setting, existing methods (e.g., DeepSynergy, HGNN, MCHNN) typically rely on a **single, globally shared embedding per entity node**. Even if their features are initialized by powerful PLMs, the downstream node embeddings remain **fixed and static across different query tuples** during inference.
>
> In contrast, CONTEXTOR explicitly contextualizes the response entity, enabling the same entity to **adapt dynamically to different query contexts**. We will revise the manuscript to explicitly scope the term "static representations" to structured relational inference and discuss contextualized LLM literature to avoid overgeneralization.
>
> ---
>
> >**Q2. Differentiation from Existing Contrastive Learning or High-order Relation Learning:** The manuscript states that existing contrastive learning predominantly focuses on pairwise relations. However, asymmetric and high-order relation extraction has been studied for years. The authors should clarify how their formulation differs substantively from these existing methods.
>
> **A2:** We respectfully believe this concern **stems from a scope misunderstanding**. Our statement specifically refers to the **contrastive learning literature**, rather than the broader literature on asymmetric or high-order relational modeling.
>
> Existing contrastive learning methods, including **CLIP-style alignment and drug–target binding models cited in our manuscript,** are predominantly formulated as **pairwise symmetric alignment objectives**. Our point was that **e**xisting contrastive learning methods are largely **limited to this pairwise relational setting**, **rather than claiming that no prior work has studied asymmetric or multi-entity relations**.
>
> Meanwhile, we fully acknowledge that prior **high-order relation learning methods** (e.g., HGSynergy) explicitly model multi-entity tuples. However, these methods generally rely on a **single symmetric tuple scoring function**, without distinguishing different inference directions (e.g., predicting the missing drug vs. the missing cell line).
>
> In contrast, CONTEXTOR explicitly reformulates high-order inference as a **directional query–response process**, where the missing-entity choice changes both the query semantics and the conditioned response representation. We will revise the manuscript to **further** **highlight this distinction**.
>
> ---
> >**Q3. Comparison with Modern LLM-Based Baselines:** The selected baselines are relatively outdated. At a minimum, the authors should compare against strong contemporary LLM-based baselines (e.g., GPT-5, Gemini 3, or other biomedical-adapted LLMs) to assess the practical competitiveness of the proposed method.
>
> **A3**: To address it, we evaluated **CONTEXTOR** against **GPT-5.2** and **Gemini 3.1** via rank-based selection (MRR). **Like recent work**, we initially excluded LLMs because standard evaluation relies on **AUROC/AUPRC**, requiring calibrated probabilities over the full search space, which LLMs cannot naturally provide without specific adaptations.
>
> As shown in **Table 1**, CONTEXTOR consistently outperforms both LLMs on **Gene Identification** and **ADR Prediction**. While GPT-5.2 shows a slight edge in Drug Discovery due to its extensive pretraining on well-documented drugs, CONTEXTOR excels at **true generalization to less-studied entities**.
>
> Critically, CONTEXTOR is built for **novel scientific discovery** via explicit relational reasoning in structured spaces, whereas LLMs are inherently bounded by textual knowledge. Furthermore, unlike API-dependent LLMs, CONTEXTOR is **lightweight, computationally efficient**, and supports **secure in-house deployment** for privacy-sensitive data.
>
> **Table 1: MRR comparison on the DGA dataset under three evaluation settings. Results are reported as mean ± standard deviation.**
>
> | Model | Drug Discovery | Gene Identification | ADR Prediction |
> | :---- | :---: | :---: | :---: |
> | Gemini-3.1 | 0.3449 ± 0.0293 | 0.2364 ± 0.0194 | 0.3478 ± 0.0532 |
> | GPT-5.2 | **0.3596 ± 0.0456** | 0.2314 ± 0.0163 | 0.3682 ± 0.0485 |
> | **CONTEXTOR** | 0.3180 ± 0.0372 | **0.2466 ± 0.0118** | **0.4092 ± 0.0402** |

---

> > ### Author Rebuttal · Reviewer_ryY6 · 2026-04-03
> >
> > Thank you for the explanation. I revise my score accordingly.

---

> > > ### Author Response · Authors · 2026-04-04
> > >
> > > We sincerely appreciate your positive feedback and valuable comments. We hope our responses above have fully addressed your questions, and we remain happy to provide any further clarifications if needed.

---

### Official Review · Reviewer_WmEM · 2026-03-12

**Soundness:** 3
**Presentation:** 2
**Significance:** 3
**Originality:** 3
**Overall Recommendation:** 5
**Confidence:** 4

**Summary:**

Inspired by the NLP domain, CONTEXTOR formulates a high-order relation extraction problem as a retrieval task given a triplet of entities. e.g., given a drug-disease entity, we attempt to infer a drug-gene or disease-gene relation.
In the biomedical domain, the semantics of each entity (e.g., drug, gene, disease) vary depending on context, but existing methods rely on static representations.
To learn a dynamic representation, authors used an ACM module parameterized with learnable parameters, allowing an entity to learn different embeddings depending on context.

**Compliance With Llm Reviewing Policy:**

Affirmed.

**Final Justification:**

My initial concerns primarily related to methodological justification and clarity of specific terminolgies. The authors’ rebuttal addressed these points satisfactorily.

**Key Questions For Authors:**

[Major]
- Could you elaborate on how graph is constructed?
- Although ACM is a key module for disentangling condition-specific embeddings, the manuscript does not include an ablation study empirically evaluating its effectiveness. Could you provide an ablation analysis to quantitatively assess the contribution of ACM on model performance?
- The term "high-order relation" seems inappropriate. Calling a "high-order relation among multiple entities" based solely on triplet interactions is misleading. How about specifying it as two-order?
- For a given entity, different embeddings may appear depending on the conditions. Can you demonstrate that an entity exhibits different embeddings depending on the context, and that this is biologically/clinically plausible?

**Limitations:**

Yes

**Strengths And Weaknesses:**

[Soundness]
- Although high-order is restricted to 2nd-order in this paper, it is limitation derived from lack of high-order benchmark.
- Although ACM is key module to disentangle condition-speicific embeddings, authors does not provide ablation study to see effectiveness of ACM.

[Presentation]
- Overall, clearly written.
- For a fair comparison, the y-axis range in figures should be aligned.
- GNN is used for initial embedding extraction from multiple entities. Lack of explanations on how graph is constructed.

[Significance]
- Robust performance gain across multiple datasets.

[Originality]
- The authors propose the ACM operator to learn context-conditioned embeddings, and they provide theoretical justification.

---

> ### Author Rebuttal · Authors · 2026-03-31
>
> We sincerely thank you for your insightful feedback and recognition of our work. We remain open to further revisions. Below, we address your comments in detail.
>
> >**Q1. Graph Construction:** Could you elaborate on how graph is constructed?
>
> **A1**: Graph construction is dataset- and backbone-dependent; we strictly follow the original protocols for fairness:
>
> * **DDC\[1\]**: Modeled as a hypergraph where each (**Drug A, Drug B, Cell Line**) triplet forms a hyperedge. Drug features are derived from GCN-based molecular graphs; cell lines from genomic/expression data.
>
> * **DMD\[2\]**: Modeled as a hypergraph with (**Drug, Microbe, Disease**) triplets as hyperedges. Node features are initialized via integrated similarity networks (e.g., GIP, disease semantics, drug chemistry).
>
> * **DGA**: Triplet-based construction. A hypergraph backbone uses (**Drug, Gene, ADR**) hyperedges with domain-specific pretrained features. Under standard GNNs, these triplets are equivalently decomposed into pairwise edges within a heterogeneous graph.
>
> ---
>
> >**Q2. Effectiveness of the ACM Module:** Although ACM is a key module for disentangling condition-specific embeddings, the manuscript does not include an ablation study empirically evaluating its effectiveness. Could you provide an ablation analysis to quantitatively assess the contribution of ACM on model performance?
>
> **A2:** We evaluate ACM via an ablation variant (**CONTEXTOR w/o ACM**) that replaces **query-driven asymmetric  modulation** with a static linear projection (same parameter scale).
>
> Removing ACM causes a consistent and significant drop across all metrics (**Table 1**), demonstrating that ACM’s query-aware modulation is essential. This confirms that static embeddings suffer from semantic entanglement, while ACM enables accurate high-order relation inference.
>
> **Table 1: Ablation study on the DGA dataset under three different evaluation settings. Results are reported as mean ± standard deviation.**
>
> | Model | Drug Discovery |  |  | Gene Identification |  |  | ADR Prediction |  |  |
> | :---- | :---: | :---: | :---: | :---: | :---: | :---: | :---: | :---: | :---: |
> |  | AUROC (%) | AUPRC (%) | MRR | AUROC (%) | AUPRC (%) | MRR | AUROC (%) | AUPRC (%) | MRR |
> | w/o ACM | 89.68 ± 1.75 | 77.83 ± 6.64 | 0.2853 ± 0.0350 | 89.25 ± 1.57 | 77.96 ± 1.76 | 0.2270 ± 0.0359 | 91.13 ± 1.59 | 79.75 ± 4.30 | 0.3845 ± 0.0227 |
> | **CONTEXTOR** | **91.57 ± 2.40** | **82.04 ± 5.71** | **0.3180 ± 0.0372** | **91.61 ± 1.55** | **84.57 ± 2.15** | **0.2466 ± 0.0118** | **92.94 ± 1.03** | **84.61 ± 2.45** | **0.4092 ± 0.0402** |
>
> ---
>
> >**Q3. Terminology of "High-order Relation":** The term "high-order relation" seems inappropriate. Calling a "high-order relation among multiple entities" based solely on triplet interactions is misleading. How about specifying it as two-order?
>
> **A3:** We respectfully argue that the term "high-order relation" is **mathematically rigorous and aligns with the standard terminology** in modern network science, whereas using "two-order" or "second-order" would introduce significant ambiguity.
>
> In the fields of network science and graph representation learning, "higher-order" canonically refers to any interaction **beyond standard pairwise (2-node) edges, explicitly including 3-node triplets \[3, 4\]**. This convention is also standard in recent biomedical hypergraph literature, where triplet associations are explicitly termed **"higher-order triple-wise associations" \[2\]**. Therefore, our use of "high-order" for ternary relations is theoretically well-grounded.
>
> ---
>
> >**Q4. Biological/Clinical Plausibility of Contextual Embeddings:** For a given entity, different embeddings may appear depending on the conditions. Can you demonstrate that an entity exhibits different embeddings depending on the context, and that this is biologically/clinically plausible?
>
> **[Figure](https://anonymous.4open.science/r/Anonymous-figure/figure.pdf)**
>
> **A4:** As shown in **Figure**, our model captures genuine biological mechanisms rather than spurious correlations. Clinically validated SCAR-inducing pairs map in extreme proximity to the SMQ representation (e.g., (Allopurinol, NOTCH4): **0.106**; (Clindamycin, HLA-B): **0.192**). Conversely, the unrelated pair (Melphalan, CTLA4) exhibits a massive **L2 distance** of **2.326**. **This aligns perfectly with clinical pharmacology, as Melphalan is known for myelosuppression rather than severe skin necrosis.** This clear geometric separation firmly supports the biological plausibility of our framework.
>
> ---
>
> **References:**
> \[1\] (Bioinformatics 2022)Multi-way relation-enhanced hypergraph representation learning for anti-cancer drug synergy prediction.
> \[2\] (IJCAI 2023)Multi-view contrastive learning hypergraph neural network for drug-microbe-disease association prediction.
> \[3\] (Physics reports 2020)Networks beyond pairwise interactions: Structure and dynamics.
> \[4\] (Science 2016)Higher-order organization of complex networks.

---

> > ### Author Rebuttal · Reviewer_WmEM · 2026-04-03
> >
> > Thank you for your response. Most of my concerns are resolved.

---

> > > ### Author Response · Authors · 2026-04-04
> > >
> > > We sincerely appreciate your positive feedback and valuable comments on our work. Please feel free to contact us if you have any further questions or require additional information.

---

### Official Review · Reviewer_ppps · 2026-03-13

**Soundness:** 3
**Presentation:** 3
**Significance:** 3
**Originality:** 3
**Overall Recommendation:** 5
**Confidence:** 4

**Summary:**

* $\textbf{Problem}$ : Existing relational learning methods rely on static entity embeddings and symmetric similarity measures, which makes it difficult to capture context-dependent and asymmetric high-order relations in biomedical systems.
* $\textbf{Method}$ : The paper addresses this by reformulating high-order relation inference as a query–response prediction task and introducing Asymmetric Conditional Modulation (ACM) to dynamically transform candidate entity embeddings based on the query context, combined with a contrastive learning objective.
* $\textbf{Conclusion}$ : The results show that using context-conditioned entity representations with contrastive query–response learning improves the modeling and prediction of sparse and combinatorial high-order biomedical relations across multiple datasets.

**Compliance With Llm Reviewing Policy:**

Affirmed.

**Final Justification:**

My main concern was regarding the general applicability of the proposed method. In particular, I wondered whether it would remain effective on other datasets and whether the improvement would also hold in hypergraph models that incorporate contextual information. The authors’ rebuttal addressed these points well. If the additional results and clarifications presented in the rebuttal are incorporated into the manuscript, they would further highlight the strengths of the work. Therefore, I lean toward acceptance.

**Key Questions For Authors:**

* It would be helpful to understand whether the proposed method generalizes well to datasets beyond biomedical benchmarks.
* It would be helpful if the authors could include experiments on standard hypergraph benchmark datasets, even if only briefly in the appendix. Although such datasets may not contain rich semantic information, showing the results would help assess whether the method suffers significant performance degradation when semantic signals are weak. This evaluation would be valuable because in real-world data it is often unclear whether meaningful semantic structures exist, and demonstrating reasonable performance on benchmark datasets would increase confidence in applying the method in such uncertain settings.
* It would also be useful to evaluate the method on approaches such as [1,2], which automatically detect and incorporate hyperedge semantics, to see whether similar performance improvements can still be observed in those settings.

[1] (NIPS 25) disentangling hyperedges through the lens of category theory

[2] (TKDE 22) HSDN: A High-Order Structural Semantic Disentangled Neural Network

**Limitations:**

yes

**Strengths And Weaknesses:**

### Soundness
* strength : Claim, methodology is clear and is well supported by experiments

### Presentation
* strength : The paper is well written, easy to follow

### Siginificance
* strength :Contrastive learning for hypergraph is relatively underexplored topic, which makes this work significant in some point.

### Originality
* strength :  The individual components themselves largely build upon existing techniques but were combined to effectively solve the problem

---

> ### Author Rebuttal · Authors · 2026-03-31
>
> We sincerely thank you for your insightful feedback and recognition of our work. Below, we address each of your comments in detail.
>
> >**Q1&Q2. Generalization on Non-Biomedical Benchmarks:** \> Does the proposed method generalize well to standard, non-biomedical hypergraph datasets where semantic signals might be weak or unclear?
>
> **A1:**  We first emphasize that high-order relational structures are particularly prevalent and scientifically meaningful for biomedical discoveries which constitute the primary motivation of CONTEXTOR. Nevertheless, to demonstrate that the proposed framework is not domain-specific, we further conducted additional experiments on two non-biomedical benchmarks, namely **Elliptic++** and **DBLP**, under the link prediction setting. These datasets differ significantly in structure and semantic richness from our original benchmarks:
>
> * **Elliptic++:** We sampled 2,000 triplets.
> * **DBLP:** We utilized all 4,339 available triplets.
>
> We compared the base HGNN model with its variant augmented by our CONTEXTOR module. As shown in **Table 1**, CONTEXTOR consistently improves performance across both datasets, yielding notable gains in both AUROC and AUPRC.
>
> **Table 1: Performance on Elliptic++ and DBLP datasets.**
>
> | Model | Elliptic++ |  | DBLP |  |
> | :---- | :---: | :---: | :---: | :---: |
> |  | AUROC (%) | AUPRC (%) | AUROC (%) | AUPRC (%) |
> | HGNN | 91.13 ± 5.15 | 82.13 ± 5.66 | 78.36 ± 12.36 | 54.87 ± 22.53 |
> | **CONTEXTOR** | **94.08 ± 2.41** | **90.25 ± 3.11** | **83.29 ± 5.03** | **60.69 ± 12.95** |
>
> These results demonstrate that our approach generalizes well beyond biomedical data and remains effective even in scenarios where semantic structures are limited or ambiguous.(can delete this sentence)
>
> ---
>
> >**Q3. Evaluation on Semantic-Aware Models:** \> Can similar performance improvements be observed when the method is applied to models that automatically disentangle hyperedge semantics, such as Natural-HNN \[1\] and HSDN \[2\]?
>
> **A3:** To explore this, we integrated our framework with **Natural-HNN \[1\]** and present the results in **Table 2**.
>
> The combination yields remarkable synergistic improvements across all tasks. Most notably, in ADR Prediction, CONTEXTOR boosts Natural-HNN's MRR from 0.4082 to 0.5129. This confirms that our contrastive alignment is orthogonal and highly complementary to models that inherently disentangle hyperedge semantics.
>
> **Table 2: Performance on the DGA dataset under three evaluation settings.**
>
> | Model | Drug Discovery |  |  | Gene Identification |  |  | ADR Prediction |  |  |
> | :---- | :---: | :---: | :---: | :---: | :---: | :---: | :---: | :---: | :---: |
> |  | AUROC (%) | AUPRC (%) | MRR | AUROC (%) | AUPRC (%) | MRR | AUROC (%) | AUPRC (%) | MRR |
> | HGNN | 84.50 ± 3.08 | 69.58 ± 10.61 | 0.2741 ± 0.0352 | 85.64 ± 3.37 | 72.45 ± 4.18 | 0.2144 ± 0.0237 | 85.46 ± 2.23 | 69.28 ± 8.02 | 0.3684 ± 0.0177 |
> | Natural-HNN \[1\] | 90.15 ± 1.41 | 78.28 ± 3.12 | 0.3179 ± 0.0312 | 89.65 ± 0.57 | 80.94 ± 1.18 | 0.2121 ± 0.0216 | 90.54 ± 0.83 | 77.54 ± 1.43 | 0.4082 ± 0.0407 |
> | HGNN \+ **CONTEXTOR** | 91.57 ± 2.40 | 82.04 ± 5.71 | 0.3180 ± 0.0372 | 91.61 ± 1.55 | 84.57 ± 2.15 | 0.2466 ± 0.0118 | 92.94 ± 1.03 | 84.61 ± 2.45 | 0.4092 ± 0.0402 |
> | Natural-HNN \+ **CONTEXTOR** | **94.59 ± 0.68** | **89.15 ± 0.83** | **0.3936 ± 0.0235** | **91.70 ± 2.01** | **84.62 ± 3.90** | **0.2749 ± 0.0218** | **95.29 ± 0.54** | **91.22 ± 0.71** | **0.5129 ± 0.0248** |
>
> **Regarding HSDN \[2\]:** While we fully acknowledge its theoretical relevance, the lack of publicly available source code precluded a direct empirical comparison during the short rebuttal period. We will include a detailed discussion of this work in our final version to contextualize our contributions further.
>
> ---
>
> **References:**
> \[1\] Lee Y, Lee J, Seo S, et al. Disentangling Hyperedges through the Lens of Category Theory\[C\]//The Thirty-ninth Annual Conference on Neural Information Processing Systems.
>
> \[2\] Hu B, Wang X, Feng Z, et al. Hsdn: A high-order structural semantic disentangled neural network\[J\]. IEEE Transactions on Knowledge and Data Engineering, 2022, 35(9): 8742-8756.

---

> > ### Author Rebuttal · Reviewer_ppps · 2026-04-01
> >
> > Most of my concerns have been addressed, and I have revised my score accordingly. The rebuttal clarifies the general applicability of the proposed approach, and incorporating these points into the appendix would help highlight the strengths of the work.

---

> > > ### Author Response · Authors · 2026-04-01
> > >
> > > Thank you for your encouraging remarks and support; these comments are highly beneficial to our work. We would be more than happy to discuss further if you have any additional questions.

---

### Decision · Program_Chairs · 2026-04-30

**Decision:**

Accept (regular)

**Comment:**

This paper proposes Contextualized High-order Contrastive Learning (CONTEXTOR), a general and plug-and-play framework that formulates high-order relation inference as a dynamic query–response process. All reviewers found the paper interesting, Although they raised some concerns, those are satisfactorily addressed during the authors' rebuttal process. This is a solid contribution to ICML.